# MMD Graph Kernel: Effective Metric Learning for Graphs via Maximum Mean Discrepancy

**Yan Sun**[1,2]   **Jicong Fan**[1,3*]

[1]School of Data Science, The Chinese University of Hong Kong, Shenzhen, China
[2]School of Computing, National University of Singapore, Singapore
[3]Shenzhen Research Institute of Big Data, Shenzhen, China
`yansun@comp.nus.edu.sg` `fanjicong@cuhk.edu.cn`

## ABSTRACT

This paper focuses on graph metric learning. First, we present a class of maximum mean discrepancy (MMD) based graph kernels, called MMD-GK. These kernels are computed by applying MMD to the node representations of two graphs with message-passing propagation. Secondly, we provide a class of deep MMD-GKs that are able to learn graph kernels and implicit graph features adaptively in an unsupervised manner. Thirdly, we propose a class of supervised deep MMD-GKs that are able to utilize label information of graphs and hence yield more discriminative metrics. Besides the algorithms, we provide theoretical analysis for the proposed methods. The proposed methods are evaluated in comparison to many baselines such as graph kernels and graph neural networks in the tasks of graph clustering and graph classification. The numerical results demonstrate the effectiveness and superiority of our methods. Our code is available at `https://github.com/yan-sun-x/MMD-Graph-Kernel`.

## 1 INTRODUCTION

Graphs, as mathematical structures, represent entities and their interrelationships. They are indispensable in various domains such as bioinformatics for representing proteins or molecules (Aittokallio & Schwikowski, 2006; Huber et al., 2007), and social networks for identifying communities (Girvan & Newman, 2002; Cuvelier & Aufaure, 2012). Graph comparison, with an explicit distance or similarity metric, has been a topic of great interest beyond merely classifying graphs. Graph kernels have emerged as a class of effective methods for measuring the similarities between graphs (Nikolentzos et al., 2021). They often recursively break down graphs into substructures—like paths (Borgwardt & Kriegel, 2005), graphlets (Shervashidze et al., 2009), walks (Vishwanathan et al., 2010), and subtrees (Shervashidze et al., 2011), and then compare these substructures between two graphs (Kriege et al., 2018).

However, early graph kernels face two notable limitations. Firstly, the substructures derived from graphs share nodes or edges, leading to overlapping features within the generated feature map, which inflates the dimensionality of the feature space and may reduce the efficiency of the kernel method (Ye et al., 2020). Secondly, these kernels often depend on manually crafted features, missing the nuanced relationships between vertices, thus not integrating higher-level vertex information into the graph feature maps. To address the first limitation, Yanardag & Vishwanathan (2015) proposed the Deep Graph Kernels (DGK) that leverage natural language processing techniques to get latent substructure representations and create a similarity matrix between these substructures for graph kernel matrix calculations. However, if there is a large number

---

[*]Corresponding author

of substructures, calculating the similarity matrix is time-consuming. The second limitation of graph kernels can be partially addressed by Graph Neural Networks (GNNs) (Kipf & Welling, 2016; Veličković et al., 2017; Gilmer et al., 2017; Hamilton et al., 2017; Xu et al., 2018; Sun et al., 2019; You et al., 2020; Sun et al., 2023), that have shown promising performance in both node-level tasks (e.g. node classification and link prediction) and graph-level tasks (e.g. graph classification). While GNNs may not capture more substructure patterns than WL kernels, they can learn complex functions of features across graph neighborhoods that can be more useful for certain prediction tasks. However, for graph-level tasks, GNNs have to use a readout function or operation (e.g. summation or averaging) to convert the nodes' representations (a matrix) of each graph to a single vector that can be used as the representation of the entire graph. The readout functions or operations are generally heuristic, potentially leading to the loss of structural information and further lowering the classification accuracy. Although there have been many efforts and algorithms for better graph representation learning such as Infograph (Sun et al., 2019), GraphCL (You et al., 2020), Lovász Principle (Sun et al., 2023), and GCKM (Wu et al., 2023), the final graph representation vector is still obtained by a readout operation on the node representations.

Rather than investigating substructure and graph-level representations, we can regard each graph as a discrete distribution or a sample from a distribution, to be more specific, all nodes in a graph are sampled from a population. Thus, the similarity between two distributions is a proxy to graph similarity or graph kernel. Without prior knowledge about how each graph behaves, we cannot rely on any parametric distribution to describe the graph. Fortunately, Maximum Mean Discrepancy (MMD) (Smola et al., 2006) provides an effective and efficient solution to measure the difference between two parameter-free distributions, though one may consider other integral probability metrics such as Wasserstein distance (Sriperumbudur et al., 2012; Panaretos & Zemel, 2019). The MMD-based approach can be applied: 1) even when the two graphs have unequal sizes (number of nodes) which is often the case in real data; 2) without relying on permutation on nodes; 3) for unsupervised settings where graph labels are missing.

In this study, we develop a novel approach to graph metric learning, **M**aximum **M**ean **D**iscrepancy-**G**raph **K**ernel (**MMD-GK**), and its extension **Deep MMD-GK**. The experimental results of graph clustering and graph classification on benchmarks demonstrate our methods surpass many baselines. Additionally, we analyze the theoretical robustness of our methods and provide practical guidance on model configurations.

## 2 PRELIMINARY

We present the following definitions for convenience.

**Definition 1.** *(Attributed Labeled Graph) An attributed labeled graph is a graph $G = (V, E, X, Y)$ with $n$ vertices , endowed with an attribute function $f : V \mapsto X \in \mathbb{R}^{n \times d}$ that assigns $d$-dimensional real-valued vectors to the vertices of the graph and with a label function $\zeta : V \mapsto Y \in \mathbb{R}^n$ that assigns labels to the vertices of the graph from a discrete set of labels. The adjacency matrix of $G$ is denoted as $A \in \{0, 1\}^{n \times n}$.*

In many settings, we represent the discrete labels as one-hot vectors, then $Y$ becomes a matrix. Note that we separate the labels from the attributes of graph, since many graph datasets (e.g. protein structure) include both attributes and labels.

**Definition 2.** *(l-Level Node Features) Given an attributed labeled graph $G$, the $l$-level features of a node aggregate all features in its $l$-order neighbors, including edges, node attributes, and labels. Mathematically, let $U := \tilde{D}^{-\frac{1}{2}} \tilde{A} \tilde{D}^{-\frac{1}{2}}$, where $\tilde{A} = A + I$ and $\tilde{D} = diag(\sum_j \tilde{A}_{1j}, \ldots, \sum_j \tilde{A}_{nj})$, all $l$-level features of a graph are $X^{(l)} = U^l X$, $Y^{(l)} = U^l Y$.*

Message propagation gives rise to $l$-level node attributes. The representations $X^{(l)}$ and $Y^{(l)}$ can better capture the local structure and node features (attributes and labels). The choice of $l$ depends on the case.

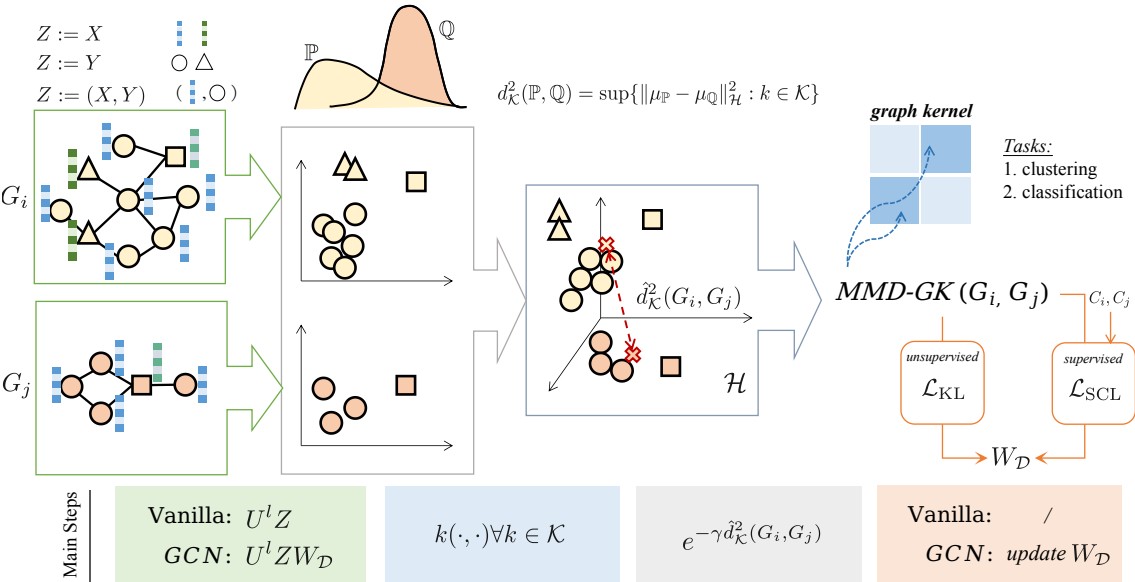

Figure 1: Architecture of MMD-Graph Kernel (**MMD-GK**) in vanilla and deep version, where the model is Graph Convolutional Networks (GCN) (Kipf & Welling, 2016). The main steps consist of embedding aggregation, kernel mapping, distance-to-similarity, and for GCN, updating parameters. **MMD-GK** can be evaluated in downstream tasks such as graph clustering and classification.

**Definition 3.** *(Maximum Mean Discrepancy, MMD) Given two distributions $\mathbb{P}$, $\mathbb{Q}$ and a kernel $k$, the square of MMD distance between $\mathbb{P}$ and $\mathbb{Q}$ is defined as*

$$d_k^2(\mathbb{P}, \mathbb{Q}) = \|\mu_{\mathbb{P}} - \mu_{\mathbb{Q}}\|_{\mathcal{H}}^2 = \mathbb{E}_{\mathbb{P}}[k(x, x')] - 2\mathbb{E}_{\mathbb{P}, \mathbb{Q}}[k(x, y)] + \mathbb{E}_{\mathbb{Q}}[k(y, y')], \tag{1}$$

*where $\mu_{\mathbb{P}}$ and $\mu_{\mathbb{Q}}$ represent the means of distributions $\mathbb{P}$ and $\mathbb{Q}$ in the reproducing kernel Hilbert space (RKHS), from which $x, x'$ and $y, y'$ are sampled respectively.*

Note that a kernel is *characteristic* if the mapping $\mathbb{P} \mapsto \mu_{\mathbb{P}}$ is injective and then $d_k(\mathbb{P}, \mathbb{Q}) = 0 \iff \mathbb{P} = \mathbb{Q}$ holds. There are many classes of characteristic kernels, such as the Gaussian kernel $\mathcal{K}_g := \{e^{-\|x-y\|_2^2/h} : x, y \in \mathbb{R}^d, h \in \mathbb{R}_+\}$. Fukumizu et al. (2009) generalized the MMD to families of unnormalized kernels.

**Definition 4.** *(Generalization of MMD) Give a family of positive definite kernels $\mathcal{K}$, the generalized MMD between $\mathbb{P}$ and $\mathbb{Q}$ over $\mathcal{K}$ is*

$$d_{\mathcal{K}}^2(\mathbb{P}, \mathbb{Q}) = \sup\{d_k^2(\mathbb{P}, \mathbb{Q}) : k \in \mathcal{K}\} = \sup\{\|\mu_{\mathbb{P}} - \mu_{\mathbb{Q}}\|_{\mathcal{H}}^2 : k \in \mathcal{K}\}. \tag{2}$$

In practice, we distinguish two distributions by *Two-Sample Test* and estimate MMD from finite samples as a distance metric. The estimator of $d_{\mathcal{K}}^2(\mathbb{P}, \mathbb{Q})$ is defined as follows.

**Definition 5.** *(Estimated MMD) With finite samples $X = \{x_i\}_{i=1}^{n_X} \sim \mathbb{P}$ and $Y = \{y_j\}_{j=1}^{n_Y} \sim \mathbb{Q}$, one estimator of $d_{\mathcal{K}}^2(\mathbb{P}, \mathbb{Q})$ with a kernel family $\mathcal{K}$ is*

$$\hat{d}_{\mathcal{K}}^2(X, Y) = \sup_{k \in \mathcal{K}} \left[ \frac{1}{n_X^2} \sum_{i,i'=1}^{n_X} k(x_i, x_{i'}) + \frac{1}{n_Y^2} \sum_{j,j'=1}^{n_Y} k(y_j, y_{j'}) - \frac{2}{n_X n_Y} \sum_{i,j=1}^{n_X, n_Y} k(x_i, y_j) \right]. \tag{3}$$

Note that the kernel family contains the same class of kernel, such as a group of Gaussian kernels with different bandwidths $\{k_h : h \in \mathbb{R}_+\}$. Compared to the single-kernel MMD, the generalized MMD has the following advantages in practice: 1) auto-search for a suitable and robust bandwidth; 2) avoid the distance collapse as a $k \to 0$ or $k \to 1$ leads to $d_k^2 \to 0$ (Fukumizu et al., 2009).

# 3 VANILLA MMD GRAPH KERNEL

## 3.1 MMD BETWEEN GRAPHS

Graphs stand apart due to their node attributes, labels, and edge structures. This distinctiveness suggests that each graph can be viewed as a manifestation of a specific distribution, especially for graphs sharing identical labels. Therefore, we consider a set of graphs as emerging from a specific distribution when they share the same label. From this perspective, we propose the concept of *graph distribution*.

**Definition 6.** *(Graph Distribution) Consider a collection of attributed labeled graphs* $\mathcal{G} := \{G_i := (V_i, E_i, X_i, Y_i)\}_{i=1}^N$. *Each graph is assigned a label through a surjective function* $c : G \mapsto C$. *The labels* $C \in \{C_1, \cdots, C_K\}$ *represent distinct distributions of node attributes and labels. For a given label* $C$, *the graph distribution* $\mathbb{P}^{l,C}$ *is defined with respect to node attribute* $X$, *label* $Y$ *and their combined feature* $(X, Y)$ *at a specific level* $l$.

We make an assumption regarding the properties of node features at various levels.

**Assumption 1.** *For any* $l \in \mathbb{R}_+$, *the* $l$-*level node features in graphs with the same label follow specific distributions. This is represented as:*

$$\{X_i^{(l)} : C_i = C\} \sim \mathbb{P}_X^{l,C}, \qquad \{Y_i^{(l)} : C_i = C\} \sim \mathbb{P}_Y^{l,C}, \qquad \{(X_i^{(l)}, Y_i^{(l)}) : C_i = C\} \sim \mathbb{P}_{(X,Y)}^{l,C}. \quad (4)$$

To quantify the differences between graph distributions, we extend Definition 4 as below.

**Definition 7.** *(MMD between Two Graphs) Given two attributed labeled graphs* $G_1$, $G_2$ *and a family of positive definite kernels* $\mathcal{K}$, *the* $l$-*level squared Maximum Mean Discrepancy between the graph distributions induced by* $G_1$, $G_2$ *over* $\mathcal{K}$ *is defined as follows*

$$d_{\mathcal{K}}^2(G_1^{(l)}, G_2^{(l)}) := d_{\mathcal{K}}^2(\mathbb{P}_Z^{l,C_1}, \mathbb{P}_Z^{l,C_2}) = \sup \left\{ \left\| \mu_{\mathbb{P}_Z^{l,C_1}} - \mu_{\mathbb{P}_Z^{l,C_2}} \right\|_{\mathcal{H}}^2 : k \in \mathcal{K} \right\}. \quad (5)$$

For $Z = (X, Y)$, the squared MMD captures the overall discrepancy between graphs by considering both node attributes and labels. The distance focuses on the discrepancy between node attributes when $Z = X$ or node labels when $Z = Y$.

## 3.2 VANILLA MMD-GK

In this section, we introduce MMD-Graph Kernel, delineated in Algorithm 2 (see Appendix C), to estimate the similarity between two graphs without information in graph labels. For convenience, we denote the graphs by $G_1 = (X_1, Y_1, U_1)$ and $G_2 = (X_2, Y_2, U_2)$, where the graph structures or the edge information are encoded by the normalized and self-looped adjacency matrices $U_1$ and $U_2$ (see Definition 2).

**Key Steps:** The model employs an embedding aggregation mechanism to iteratively update these node representations till $L$ levels. Specifically, at each level $l$, the node representations are updated according to the normalized adjacency matrix. Subsequently, the model utilizes MMD to quantify the dissimilarity between two sets of node representations, $Z_1^{(l)}$ with a size of $n_1$ and $Z_2^{(l)}$ with a size of $n_2$. The squared

MMD is estimated using a family of Gaussian kernels $\mathcal{K}$, described as follows:

$$\hat{d}_{\mathcal{K}}^2(Z_1^{(l)}, Z_2^{(l)}) = \sup_{k \in \mathcal{K}} \left[ \frac{1}{n_1^2} \sum_{i,i'=1}^{n_1} k(z_{i,1}, z_{i',1}) + \frac{1}{n_2^2} \sum_{j,j'=1}^{n_2} k(z_{j,2}, z_{j',2}) - \frac{2}{n_1 n_2} \sum_{i,j=1}^{n_1,n_2} k(z_{i,1}, z_{j,2}) \right]. \quad (6)$$

Note that $\hat{d}_{\mathcal{K}}$ is a distance metric that satisfies the metric space axioms (e.g. triangle inequality). By default, a Gaussian kernel parameterized by a bandwidth $h > 0$ is employed for this purpose. The kernel function measures the similarity between two node representations $z_i$ and $z_j$, and is defined as $k(z_i, z_j) = \exp\left(-\frac{\|z_i - z_j\|^2}{h_k}\right)$. Finally, we obtain a graph kernel $s$ as

$$s_L(G_1, G_2) = \exp\left(-\gamma \cdot \hat{d}_{\mathcal{K}}^2(Z_1^{(L)}, Z_2^{(L)})\right), \quad (7)$$

where $\gamma > 0$ is a hyperparameter. This kernel serves as a metric for comparing and quantifying the similarity between different graphs. Since $\hat{d}_{\mathcal{K}}$ is a distance metric in Hilbert space and the exponential function is positive and monotonic increasing, $s_L$ is a strictly positive-definite kernel.

### 3.3 ROBUSTNESS ANALYSIS

We conduct the robustness analysis of the vanilla MMD-GK. As mentioned by O'Bray et al. (2021), a suitable metric such as MMD should be robust to small perturbations. We need to find an ideal upper-bound function of the amplitude of the perturbation taking effects in the metric. Let $\Delta_X$, $\Delta_Y$, and $\Delta_A$ be some perturbations on node attributes, node labels, and graph structure and denote the perturbed graph as $\tilde{G} = (A + \Delta_A, X + \Delta_X, Y + \Delta_Y)$. To simply notations, we let $Z = (X, Y)$ and $\Delta_Z = (\Delta_X, \Delta_Y)$.

**Theorem 1.** *For any two graphs $G_1, G_2$, without loss of generality, suppose $n_1 = n_2 = n$ and the minimum node degrees of $G_1, G_2$ are both $\alpha$. Suppose $\|A_i\|_2 \leq \beta_A$, $\|X_i\|_2 \leq \beta_X$, $\|Y_i\|_2 \leq \beta_Y$, and $\|Z_i\|_F \leq \eta$, $i = 1, 2$. Denote the effects of structural perturbation as $\kappa = \min(1^\top \Delta_{A_i})$ and $\Delta_{D_i} = diag(1^\top(\tilde{A}_i + \Delta_{A_i}))^{\frac{1}{2}} diag(1^\top A_i)^{-\frac{1}{2}} - I$ for $i = 1, 2$. Then the following inequality for **MMD-GK** holds*

$$\hat{d}_{\mathcal{K}}^2(\tilde{G}_1^{(l)}, \tilde{G}_2^{(l)}) \leq \hat{d}_{\mathcal{K}}^2(G_1^{(l)}, G_2^{(l)}) + \left(\frac{4}{h^2} + \frac{2\epsilon^4}{h^4}\right)\left(2\Delta_{G_1}^4 + 2\Delta_{G_2}^4 + n(\Delta_{G_1} + \Delta_{G_2})^2 + \frac{\epsilon}{\sqrt{n}}(\Delta_{G_1} + \Delta_{G_2})\right)$$

*where $\Delta_{G_i^{(l)}} = 2\eta\left(\frac{4\beta_A \|\Delta_{D_i}\|_2 + \|\Delta_{A_i}\|_2}{1+\alpha}\right)^l + \left(\frac{2\beta_A + \|\Delta_{A_i}\|_2}{1+\alpha+\kappa}\right)^l \|\Delta_{Z_i}\|_F$ for $i = 1, 2$, $\epsilon = 2(1+\alpha)^{-l}(1 + \beta_A)^l(\beta_X + \beta_Y)$, and $h$ is the optimal kernel bandwidth among family $\mathcal{K}$.*

In the theorem, the norm assumptions by $\beta_A$, $\beta_X$, $\beta_Y$, and $\eta$ are quite standard. $\Delta_{D_i}$ quantifies the perturbation on node degrees and is zero if $\Delta_{A_i} = 0$. The bound is bi-quadratically with $\eta$. Due to space limitation and the presence of Theorem 2, we defer the detailed discussion to Appendix C.1.

## 4 DEEP MMD-GRAPH KERNEL

We propose an extension termed Deep MMD-Graph Kernel (Deep MMD-GK), which enhances the model's flexibility and expressiveness. In Algorithm 1, it introduces trainable parameters $\mathcal{W} := \{W^{(i)}\}_{i=1}^L$ into the embedding aggregation step, allowing the model to learn optimal transformations of node representations based on the graph structure. The loss function of Deep MMD-GK depends on the availability of graph labels. To train the model in a supervised manner, we adopt the supervised version of InfoNEC loss function.

---

**Algorithm 1:** Deep MMD-Graph Kernel

---

**Input** : $\mathcal{G} := \{G_i\}_{i=1}^N$, $L$, $\mathcal{K}$ (with parameters $\{h_i\}_{i=1}^{|\mathcal{K}|}$), $\gamma$, $\mathcal{L}$, $\{C_i\}_{i=1}^N$ ($\emptyset$ if unsupervised), $\sigma$.

**Output:** Similarities $\{s(G_i, G_j)^L\}_{i<j}$ and model parameters $\{W^{(i)}\}_{i=1}^L$.

1 Initialize $\{W^{(i)}\}_{i=1}^L$ and let $Z_i^{(0)} \leftarrow (X_i, Y_i)$, for $i = 1, \cdots, N$.

2 **repeat**

3     **for** $l = 1$ *to* $L$ **do**

4         **for** $i = 1$ *to* $N$ **do**

5             $Z_i^{(l)} \leftarrow \sigma\left(U_i Z_i^{(l-1)} W^{(l)}\right)$    `// Embedding Propagation with Parameters` $W^{(l)}$

6         **end for**

7     **end for**

8     **for** $i = 1$ *to* $N - 1$ **do**

9         **for** $j = i$ *to* $N$ **do**

10             Compute $\hat{d}_{\mathcal{K}}^2(Z_i^{(L)}, Z_j^{(L)})$ using Eq. 6 with $Z_i^{(L)}, Z_j^{(L)}, \mathcal{K}$.    `// MMD`

11             $s(G_i, G_j)^L \leftarrow \exp\left(-\gamma \hat{d}_{\mathcal{K}}^2(Z_i^{(L)}, Z_j^{(L)})\right)$    `// Distance-to-Similarity`

12         **end for**

13     **end for**

14     $\{W^{(i)}\}_{i=1}^L \leftarrow$ Backpropagation$(\mathcal{L}, \{s(G_i, G_j)^L\}_{i<j}, \{C_i\}_{i=1}^N)$ `// Update Parameters`

15 **until** *stop criterion is reached*

16 **return** $\{s(G_i, G_j)^L\}_{i<j}$ and $\{W^{(i)}\}_{i=1}^L$

---

Specifically, inspired by (Oord et al., 2018), we use the following **Supervised Contrastive Loss (SCL)**:

$$\mathcal{L}_{\text{SCL}}(\mathcal{W}) = -\sum_{i \neq j} \mathbb{I}_{C_i = C_j} \left( \log s_{ij} - \log \left[ \sum_k \mathbb{I}_{[C_i = C_k, i \neq k]} s_{ik} + \alpha \sum_k \mathbb{I}_{[C_i \neq C_i]} s_{ik} \right] \right) \qquad (8)$$

When all training graphs are unlabeled, we can still learn Deep MMD-GK using the following two loss functions. The first one is an **Unsupervised Contrastive Loss (UCL)**, which is a modification of SCL where the negative samples come from the least similar pairs in each epoch, i.e.,

$$\mathcal{L}_{\text{UCL}}(\mathcal{W}) = -\sum_{i \neq j} \mathbb{I}_{s_{ij} \in S_+} \left( \log s_{ij} - \log \left[ \sum_k \mathbb{I}_{[s_{ik} \in S_+, i \neq k]} s_{ik} + \alpha \sum_k \mathbb{I}_{[s_{ik} \in S_-]} s_{ik} \right] \right) \qquad (9)$$

where the sets $S_+$ and $S_-$ contain the most and the least similar pairs, respectively. For each batch of training data $B$, there is a hyparameter $\lambda$ balancing the sizes as $|S_-| = |S_+| = \lfloor \lambda |B| \rfloor$. The primary benefit of UCL is that it facilitates the learning of discriminative features from a set of graphs. The second choice of unsupervised loss is the following **KL Divergence** adapted from (Xie et al., 2016):

$$\mathcal{L}_{\text{KL}}(\mathcal{W}) = \mathbb{KL}(s, s') \qquad (10)$$

where $s'_{ij} = \frac{s_{ij}^2/f_j}{\sum_{j'} s_{ij'}^2/f_{j'}}$ and $f_j = \sum_i s_{ij}$ are soft cluster frequencies. By minimizing the KL Divergence, the model learns to adjust its embeddings to better reflect the natural cluster property of the data.

### 4.1 ROBUSTNESS ANALYSIS

Similar to Section 3.3, we analyze the robustness of our Deep MMD-GK. For Deep MMD-GK, the robustness not only shows the sensitivity of the kernel to noise or perturbation but also implies the generalization ability to new graph pairs. For the detailed proof, please refer to Appendix D.2.

**Theorem 2.** *For any two graphs $G_1, G_2$, without loss of generality, suppose $n_1 = n_2 = n$ and the minimum node degrees of $G_1, G_2$ are both $\alpha$. Suppose for $i = 1, 2$, $\|A_i\|_2 \le \beta_A, \|X_i\|_2 \le \beta_X, \|Y_i\|_2 \le \beta_Y, \|Z_i\|_F \le \eta$, $\|W^{(l)}\|_2 \le \beta_{W^{(l)}}, l = 1, 2, \ldots, L$, and the activation function $\sigma$ is $\rho$-Lipschitz continuous. Denote the effects of structural perturbation as $\kappa = \min(1^\top \Delta_{A_i})$. Then the inequality for **Deep MMD-GK** holds:*

$$\hat{d}^2_{\mathcal{K}}(\tilde{G}_1^{(l)}, \tilde{G}_2^{(l)}) \le \hat{d}^2_{\mathcal{K}}(G_1^{(l)}, G_2^{(l)}) + \left(\frac{4}{h^2} + \frac{2\epsilon^4}{h^4}\right)\left(2\Delta_{G_1}^4 + 2\Delta_{G_2}^4 + n(\Delta_{G_1} + \Delta_{G_2})^2 + \frac{\epsilon}{\sqrt{n}}(\Delta_{G_1} + \Delta_{G_2})\right)$$

*where $\Delta_{G.} = \left(\frac{\rho}{1+\alpha+\kappa}\right)^l \prod_{i=1}^l \beta_{W^{(i)}}\left((\beta_A + \|\Delta_{A.}\|_2)^l \|\Delta_Z\|_F + \eta \sum_{j=0}^{l-1} \beta_A^j \left(\sqrt{(\beta_A + \|\Delta_{A.}\|_2)^2 - \beta_A^2}\right)^{l-j}\right)$, $\epsilon = 2(1+\alpha)^{-l}(1+\beta_A)^l(\beta_X + \beta_Y)$, and $h$ is the optimal kernel bandwidth among kernel family $\mathcal{K}$.*

The theorem implies how the perturbed distance metric deviates from the original: *1) the deviation in Deep MMD-GK grows bi-quadratically with $\|\Delta_Z\|_F$; 2) the deviation would be affected by $\|\Delta_A\|_2$ growing exponentially with power of $l$; 3) when $\Delta_A$ occurs, it also amplifies the perturbation $\|\Delta_Z\|_F$ ; 4) $\Delta_{G_1}$ and $\Delta_{G_2}$ take an interaction effect in contaminating Deep MMD-GK; 5) the deviation is the most significant when $n = \frac{\epsilon^3}{8(\Delta_{G_1}+\Delta_{G_2})^3}$. 6) a larger $h$ implies a smoother kernel that may be more robust to perturbations.*

Besides, the bound can be used to evaluate the generalization ability of Deep MMD-GK. Specifically, given a test pair $\tilde{G}_1, \tilde{G}_2$, we find their closest graphs, denoted by $G_1, G_2$ in $\mathcal{G}$. Then according to Theorem 2, $\hat{d}^2_{\mathcal{K}}(\tilde{G}_1^{(l)}, \tilde{G}_2^{(l)}) \le \hat{d}^2_{\mathcal{K}}(G_1^{(l)}, G_2^{(l)}) + \bar{\Delta}$, where $\bar{\Delta}$ is small provided that $G_1, G_2$ are similar to $\tilde{G}_1, \tilde{G}_2$ in terms of the adjacency matrices and node features. Note that the robustness-based generalization framework (Xu & Mannor, 2012) is not applicable to our method, where the training data (graph pairs) are non-i.i.d.

## 4.2 GENERALIZATION ERROR BOUND OF SUPERVISED LEARNING

Suppose $\mathcal{W}$ are known, Algorithm 1 with SCL is guaranteed with a uniform stability parameter $\omega$ (see Theorem 5 of Appendix F). Let $\mathcal{E}[\ell(\mathcal{W})]$ be the empirical risk, defined by Eq. 8 ($\mathcal{E}[\ell(\mathcal{W})] = \frac{1}{N(N-1)}\mathcal{L}_{\text{SCL}}(\mathcal{W})$) and let $\mathbb{E}[\ell(\mathcal{W})]$ be the true risk. According to (Bousquet & Elisseeff, 2002; Feldman & Vondrak, 2019), the estimation error of supervised loss has a high-probability generalization bound that for some constant $c$ and any $\delta \in (0, 1)$, the following inequality holds:

$$\mathbf{Pr}\left[\left|\mathbb{E}[\ell(\mathcal{W})] - \mathcal{E}[\ell(\mathcal{W})]\right| \ge c\left(\omega \log(N)\log(N/\delta) + \sqrt{\frac{\log(1/\delta)}{N}}\right)\right] \le \delta. \tag{11}$$

We see that a smaller $\omega$ (higher stability) leads to a tighter generalization error bound. Note that the objective is highly nonconvex and the optimal $\mathcal{W}$ are hard to obtain, thus we will work in the future to find the explicit form of the generalization bound.

## 4.3 COMPUTATIONAL COMPLEXITY ANALYSIS

The theoretical computational complexity of vanilla and Deep MMD-GKs are shown by the following two theorems (proved in Appendix G).

**Theorem 3.** *The MMD-GK using kernels $\mathcal{K}$ on a pair of graphs $G$ and $G'$, with $L$-level of $d$-dimensional node features, can be computed in $\mathcal{O}(Lmd + \kappa n^2 d)$, where $n$ is the number of nodes, $m$ is the number of edges, and $\kappa$ is the size of the kernel family, i.e. $\kappa = |\mathcal{K}|$. For $N$ graphs, all pairwise MMD-GKs are computed in $\mathcal{O}(NLmd + N^2\kappa n^2 d)$.*

Table 1: Comparison of the computational complexity of selected graph kernels regarding support for node-labeled and node-attributed graphs. $k$ is the size of the largest subgraph considered. $h$ is the maximum distance between the root of the neighborhood subgraph/subtree pattern and its nodes.

| Graph Kernel | Node Labels | Node Attributes | Complexity |
|---|---|---|---|
| Graphlet | ✗ | ✗ | $\mathcal{O}(n^k)$ |
| Shortest Path | ✓ | ✓ | $\mathcal{O}(n^4)$ |
| Weisfeiler-Lehman Subtree (Shervashidze et al., 2011) | ✓ | ✗ | $\mathcal{O}(hm + hn)$ |
| Wasserstein Weisfeiler-Lehman (Togninalli et al., 2019) | ✓ | ✓ | $\mathcal{O}(hm + n^3 \log(n))$ |
| MMD-GK | ✓ | ✓ | $\mathcal{O}(Lmd + \kappa n^2 d)$ |
| Deep MMD-GK | ✓ | ✓ | $\mathcal{O}(Lnd^2 + Lmd + \kappa n^2 d)$ |

**Theorem 4.** *Suppose the widths of the hidden layers of the neural network are $\mathcal{O}(d)$. The Deep MMD-GK using kernels $\mathcal{K}$ on a pair of graphs $G$ and $G'$, with $L$-level of $d$-dimensional node features, can be computed in $\mathcal{O}(Lnd^2 + Lmd + \kappa n^2 d)$, where $n$ is the number of nodes, $m$ is the number of edges, and $\kappa$ is the size of the kernel family, i.e. $\kappa = |\mathcal{K}|$. For $N$ graphs, all pairwise Deep MMD-GKs are computed in $\mathcal{O}(NLnd^2 + NLmd + N^2\kappa n^2 d)$.*

Table 1 shows that our MMD-GK and Deep MMD-GK are competitive with other kernels in terms of computational complexity. Generally, we set a $\kappa$ s.t. $\kappa d < n$, so the MMD-GK is better in complexity than the Wasserstein Weisfeiler-Lehman graph kernel, when holding $L$ equal to $h$. Note that the Weisfeiler-Lehman Subtree graph kernels scale linearly with the number of nodes; therefore, this method is faster than our kernels. The time cost comparison is reported in Appendix G.

## 5 RELATED WORK

The contents are in Appendix A.

## 6 EXPERIMENTS

The proposed methods are compared with state-of-the-art graph kernels and GNNs in both graph classification and graph clustering, on five benchmarks including DHFR, BZR, MUTAG, PTC_FM, and PROTEINS from (Kersting et al., 2016). Please see Appendix H for more details on configurations. We tuned the hyperparameters of all methods and report their best performances (detailed in Appendix K).

**Classification Accuracy** Table 2 shows that our methods always score high in graph classification. The vanilla MMD-GK, while not always achieving top scores, is competitive, especially on DHFR and MUTAG. The unsupervised and supervised Deep MMD-GKs generally demonstrate superior performance. They not only outperform traditional graph kernel methods but also surpass some of the advanced deep learning-based models on BZR, MUTAG, and PTC_FM.

**Clustering Scores** Table 3 presents the clustering accuracy (ACC), Normalized Mutual Information (NMI), and Adjusted Rand Index (ARI). Our MMD-GK and Deep MMD-GK, show significant improvement over graph kernels and GNN-based methods InfoGraph (Sun et al., 2019) and GraphCL (You et al., 2020) followed by spectral clustering. Particularly, the Deep MMD-GK (unsupervised) exhibits consistently high performance across all datasets. On BZR, it achieves 0.757 in NMI and 0.809 in ARI while the baselines failed to provide meaningful clustering results in terms of NMI and ARI.

**Practical Insights** *1) MMD-GK:* In general, as $l$ increases, the performance tends to improve at first, reaching a peak. Beyond a certain value of $l$, the performance starts to decline or plateau. For all three

Table 2: Graph classification accuracy. Deep MMD-GK (unsupervised) is trained by $\mathcal{L}_{KL}$ or $\mathcal{L}_{UCL}$, of which the better is showed. Deep MMD-GK (supervised) is trained by $\mathcal{L}_{SCL}$. *The best score on each dataset is underlined. We highlight our methods that exceed the baseline by using bold text.*

| Method | BZR | DHFR | MUTAG | PTC_FM | PROTEINS |
|---|---|---|---|---|---|
| Graphlet Kernel | $0.788 \pm 0.005$ | $0.609 \pm 0.001$ | $0.665 \pm 0.009$ | $0.610 \pm 0.002$ | $0.714 \pm 0.003$ |
| Shortest Path | $0.788 \pm 0.033$ | $0.603 \pm 0.006$ | $0.782 \pm 0.041$ | $0.639 \pm 0.022$ | $0.758 \pm 0.006$ |
| Weisfeiler-Lehman | $0.785 \pm 0.006$ | $0.683 \pm 0.111$ | $0.788 \pm 0.048$ | $0.645 \pm 0.030$ | $0.755 \pm 0.002$ |
| Wasserstein WL (Togninalli et al., 2019) | $0.785 \pm 0.006$ | $0.793 \pm 0.029$ | $0.788 \pm 0.048$ | $0.663 \pm 0.030$ | $0.755 \pm 0.002$ |
| DGCNN (Zhang et al., 2018) | $0.844 \pm 0.002$ | $0.793 \pm 0.056$ | $0.873 \pm 0.015$ | $0.603 \pm 0.067$ | $0.772 \pm 0.009$ |
| DGK (Yanardag & Vishwanathan, 2015) | $0.831 \pm 0.005$ | $0.641 \pm 0.009$ | $0.874 \pm 0.003$ | $0.645 \pm 0.008$ | $0.726 \pm 0.005$ |
| GNTK (Du et al., 2019) | $0.836 \pm 0.029$ | $0.735 \pm 0.007$ | $0.890 \pm 0.009$ | $0.639 \pm 0.001$ | $0.756 \pm 0.042$ |
| DeepMap (Ye et al., 2020) | $0.890 \pm 0.048$ | $\underline{0.852 \pm 0.022}$ | $0.895 \pm 0.031$ | $0.652 \pm 0.056$ | $0.762 \pm 0.029$ |
| MMD-GK | $0.788 \pm 0.005$ | $0.828 \pm 0.034$ | $0.819 \pm 0.073$ | $0.665 \pm 0.009$ | $0.750 \pm 0.015$ |
| Deep MMD-GK (unsupervised) | $\underline{\mathbf{0.910 \pm 0.111}}$ | $0.836 \pm 0.042$ | $\mathbf{0.910 \pm 0.026}$ | $\mathbf{0.667 \pm 0.028}$ | $0.754 \pm 0.023$ |
| Deep MMD-GK (supervised) | $\underline{\mathbf{0.910 \pm 0.111}}$ | $0.848 \pm 0.091$ | $\underline{\mathbf{0.915 \pm 0.065}}$ | $\underline{\mathbf{0.668 \pm 0.031}}$ | $\underline{\mathbf{0.776 \pm 0.025}}$ |

measures (ACC, NMI, and ARI) in the graph clustering task, Figures 4, 5 and 6 in Appendix I illustrate that there's a notable fluctuation in performance concerning changes in $l$. The sensitivity to $l$ indicates that the method captures different levels of structural and attribute information from the graph as $l$ changes. *2) Deep MMD-GK:* When graph labels are unavailable, unsupervised Deep MMD-GKs outperform vanilla MMD-GK. In addition, unsupervised Deep MMD-GKs trained by the loss $\mathcal{L}_{KL}$ at a higher $l$ (e.g. 4) is more effective and stable than $\mathcal{L}_{UCL}$ across all benchmarks.

## 7 CONCLUSION

We have proposed MMD-based graph kernels for calculating the similarity between graphs. MMD-GK and Deep MMD-GK directly compare the intrinsic distribution between two graphs via Maximum Mean Discrepancy. The key steps of our approaches are embedding aggregation, distance-to-similarity, and for the deep model, updating parameters in a supervised or unsupervised manner. The empirical results show that our models outperform the classical graph kernel and even GNN methods significantly in the tasks of graph clustering and classification. We also contribute theoretically by providing upper bounds for MMD-GK and Deep MMD-GK in robustness analysis and analyzing the generalization error bound for supervised learning. Future work may focus on improving the scalability of our methods and providing better generalization error bounds.

Table 3: Graph clustering results. *The best score on each dataset is underlined. We highlight our methods that exceed the baseline by using bold text.*

| Method | Metric | BZR | DHFR | MUTAG | PTC_FM | PROTEINS |
|---|---|---|---|---|---|---|
| Graphlet Kernel | ACC | 0.753 | 0.567 | 0.771 | 0.928 | 0.622 |
| | NMI | 0.020 | 0.003 | 0.143 | 0.036 | 0.053 |
| | ARI | 0.079 | 0.006 | 0.234 | 0.039 | 0.027 |
| Shortest Path | ACC | 0.654 | 0.679 | 0.676 | $\underline{0.962}$ | 0.604 |
| | NMI | 0.005 | 0.004 | 0.211 | 0.032 | 0.027 |
| | ARI | -0.019 | -0.007 | -0.318 | 0.021 | 0.041 |
| Weisfeiler-Lehman | ACC | 0.767 | 0.666 | 0.548 | 0.928 | 0.582 |
| | NMI | 0.014 | 0.007 | 0.136 | 0.036 | 0.021 |
| | ARI | 0.035 | 0.000 | 0.176 | 0.039 | 0.026 |
| InfoGraph | ACC | 0.735 | 0.658 | 0.726 | 0.620 | 0.732 |
| | NMI | 0.036 | 0.032 | 0.287 | 0.021 | 0.132 |
| | ARI | 0.050 | 0.005 | 0.199 | 0.046 | 0.124 |
| GraphCL | ACC | 0.729 | 0.652 | 0.732 | 0.621 | 0.728 |
| | NMI | 0.019 | 0.040 | 0.322 | 0.021 | 0.140 |
| | ARI | 0.035 | 0.003 | 0.234 | 0.034 | 0.115 |
| MMD-GK | ACC | **0.807** | 0.614 | 0.707 | 0.633 | 0.730 |
| | NMI | **0.095** | **0.042** | 0.141 | 0.034 | **0.144** |
| | ARI | **0.232** | 0.004 | 0.168 | **0.060** | **0.207** |
| Deep MMD-GK (unsupervised) | ACC | $\underline{\mathbf{0.952}}$ | $\underline{\mathbf{0.689}}$ | $\underline{\mathbf{0.846}}$ | 0.647 | $\underline{\mathbf{0.752}}$ |
| | NMI | $\underline{\mathbf{0.757}}$ | $\underline{\mathbf{0.182}}$ | $\underline{\mathbf{0.468}}$ | $\underline{\mathbf{0.076}}$ | $\underline{\mathbf{0.333}}$ |
| | ARI | $\underline{\mathbf{0.809}}$ | $\underline{\mathbf{0.137}}$ | $\underline{\mathbf{0.476}}$ | $\underline{\mathbf{0.066}}$ | $\underline{\mathbf{0.345}}$ |

ACKNOWLEDGEMENTS

This work was supported by the National Natural Science Foundation of China under Grant No.62376236, the General Program JCYJ202103241302008022 of Shenzhen Fundamental Research, the research funding T00120210002 of Shenzhen Research Institute of Big Data, and the funding UDF01001770 of The Chinese University of Hong Kong, Shenzhen.

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

## A    RELATED WORK

### A.1    GRAPH SIMILARITY

Graph Similarity is a fundamental concept in graph theory and deals with the problem of determining how alike two or more graphs are. A foundational notion here is Graph Isomorphism (Babai, 2016). Two graphs are deemed isomorphic if they contain a one-to-one correspondence between their vertices, such that the adjacency between vertices is preserved. Delving into Graph Kernels, several kernels exist with different methods of operation. The Shortest Path Kernel (Borgwardt & Kriegel, 2005; Hermansson et al., 2015), for instance, examines all pairs of shortest paths in two graphs and identifies paths of the same length. Graphlet Kernels (Shervashidze et al., 2009; Vacic et al., 2010) work by counting the number of small subgraphs, known as graphlets, that are isomorphic between two larger graphs. Considering attributed graphs, Kriege and Mutzel Kriege & Mutzel (2012) proposed the subgraph matching kernel which is based on structure-preserving bijections between subgraphs. Another interesting method is the Random Walk Kernel (Gärtner et al., 2003; Sugiyama & Borgwardt, 2015) which gauges similarity based on the number of random walks two graphs can share, effectively calculating the likelihood that a random walk in one graph mirrors a walk in the other. The Weisfeiler-Lehman Kernel (Shervashidze et al., 2011) employs the Weisfeiler-Lehman isomorphism test (Leman & Weisfeiler, 1968). This involves a continuous refinement of vertex labels based on the labels of neighboring vertices, and the kernel then counts the common labels between the two graphs. Besides, the idea to identify the best possible matching has been used in graph kernels, from early optimal assignment kernel (Fröhlich et al., 2005), to many variants such as Weisfeiler-Lehman optimal assignment kernel (Kriege et al., 2016). A notable trend in recent research has been the modification of Weisfeiler-Lehman (WL) kernels using Wasserstein distances (Togninalli et al., 2019; Chen et al., 2022; Schulz et al., 2022). Togninalli et al. (2019) integrated the Wasserstein distance into the WL framework, allowing for a more refined comparison of graphs, particularly those with continuous node attributes. Chen et al. (2022)

proposed a novel concept of WL distance, a polynomial time computable metric that is sensitive to more subtle graph differences than traditional WL methods.

With the advent and dominance of deep learning, efforts have been made to harness neural networks for deriving learned representations of graphs. The likeness between two graphs can then be determined by comparing these overarching representations. A notable subtype of graph neural networks is the Graph Convolutional Networks (GCNs) (Kipf & Welling, 2016) which employ a form of convolution to assimilate information from neighboring nodes. Another method, Deep Graph Kernels (DGK) (Yanardag & Vishwanathan, 2015), combines deep learning's expressive power with the interpretability of traditional graph kernels by utilizing deep learning for graph representation and subsequently applying traditional kernel methods to these representations. Du et al. (2019) presented Graph Neural Tangent Kernels (GNTK) which correspond to infinitely wide multi-layer GNNs trained by gradient descent. Theoretically, GNTKs provably learn a class of smooth functions on graphs. Ye et al. (2020) proposed a framework called DeepMap to learn deep representations for graph feature maps. DeepMap generates aligned vertex sequences and builds each vertex's receptive field to learn a dense and low-dimensional vector that captures complex high-order interactions in a vertex neighborhood.

## A.2 MAXIMUM MEAN DISCREPANCY

At its core, MMD (Smola et al., 2006) measures the distance between two distributions, providing a lens through which we can compare them. It leverages the power of Reproducing Kernel Hilbert Space (RKHS), enabling a non-parametric comparison between distributions. The distance is computed by taking samples from two distributions and evaluating their difference in the RKHS. With the aid of a kernel function (often Gaussian), MMD evaluates how close the mean embeddings of these samples are in the RKHS. If the embeddings are similar, it's an indication that the two samples likely come from the same distribution. Taking advantage of it, Gretton et al. (2012) demonstrated that MMD forms the basis for a kernel-based two-sample test. In situations where traditional methods, such as the Kolmogorov-Smirnov or Mann-Whitney U test, may struggle (especially in high-dimensional settings), the MMD-based test can offer greater power. Combining with a graph kernel, Borgwardt (2007) obtain the two-sample test for sets of graphs. It is important to note that, Borgwardt (2007) just proposed to estimate the MMD for two-sample test with existing graph kernels, rather than to design a graph kernel with an MMD method. When it comes to designing a novel graph kernel, a recent work focusing on speeding up Graphlet kernels, incorporated an MMD metric when showing the effectiveness of their kernel design (Ghanem et al., 2021). However, the kernel itself does not require calculating MMD. Instead, MMD is only used to demonstrate the classification power of their embeddings.

Some recent studies have integrated MMD into the realm of GNNs. One key application is in the comparison of graph-structured data to evaluate generated graphs (Dai et al., 2020; Chen et al., 2021), for instance, when determining if two graphs exhibit similar structural properties or when comparing the node embeddings generated by GNNs from different graphs. MMD also aids in the training of GNNs (Roncoli et al., 2023), ensuring that the distribution of the generated node embeddings aligns well with the target distribution. This can be especially crucial in semi-supervised settings where labeled data is sparse but there's a need to ensure that the graph's overall structure is well captured. Recently, O'Bray et al. (2021) critically evaluated the use of MMD in graph generative model comparison, proposing practical recommendations for its effective application.

In our approach, we introduce a novel graph kernel that uniquely integrates the concept of Maximum Mean Discrepancy (MMD) with Graph Neural Networks (GNNs), differing from existing efforts that either apply MMD for two-sample tests using traditional graph kernels or use MMD to evaluate the performance of GNNs without directly incorporating it into the kernel design.

## B  EXAMPLE ILLUSTRATION

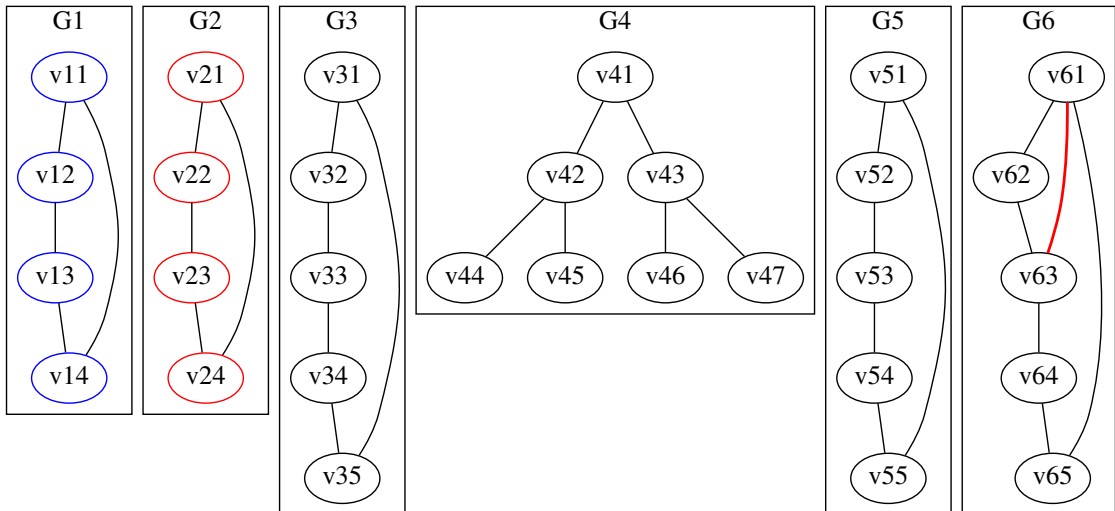

Figure 2: Graph Illustration: common scenario and counter-examples

To illustrate the concept of Maximum Mean Discrepancy (MMD) between graphs and its behavior, we present some toy examples. In each example, we consider a pair of attributed labeled graphs and estimate the supremum over MMDs ($\hat{d}_{\mathcal{K}}$) using a class of Gaussian kernels with different bandwidths.

- **Common Scenario:** Consider two graphs, $G_1$ and $G_2$, with the same graph structure but different node features. Both graphs have the same edges ($E_1 = E_2$), but the attributes of their nodes are different ($X_1 \neq X_2$). When estimating the MMD between their node attributes $X_1$ and $X_2$, the supremum over the Gaussian kernel may cause $\hat{d}_{\mathcal{K}}$ to be relatively large, indicating a notable discrepancy between the node attributes.

The MMD measures the discrepancy between the distributions of node features and labels of two graphs at the $l$-level, using the kernel family $\mathcal{K}$ to compare individual node features. If two graphs are identical, their MMD will be zero, as there will be no difference in the node characteristics between the graphs. However, the estimated MMD may appear very close to zero even when the two graphs are actually distinct, i.e. $\hat{d}_{\mathcal{K}} \approx 0$ when $X_1 \neq X_2$ or $Y_1 \neq Y_2$ or $E_1 \neq E_2$. This is because the limited number of nodes and edges in the graphs might not fully capture all the differences between them, leading to potentially underestimated MMD values. Let's consider the counter-examples:

- **Counter-example 1:** Consider two graphs, $G_3$ and $G_4$, where $G_3$ contains a cycle and $G_4$ contains a tree structure. Although they have different graph structures ($E_3 \neq E_4$), when estimating the MMD between their node features $X_3$ and $X_4$, the supremum over the Gaussian kernel with a very large bandwidth may lead to a small $\hat{d}_{\mathcal{K}}$ due to the smoothing effect. This can mistakenly suggest that $G_3$ and $G_4$ are similar in terms of their node features.

- **Counter-example 2:** Suppose we have two attributed labeled graphs, $G_5$ and $G_6$, with the same node features ($X_5 = X_6$) and almost identical labels ($Y_5 \approx Y_6$). Additionally, the edge structures of the two graphs are also very similar, but there exist a few edges that differ between them ($E_5 \approx E_6$

with some minor differences). Now, let's consider the MMD at different levels $l$ to compare the distributions of node features, labels, and edge structures:

1. At a low level, when $l = 1$: In this case, the MMDs $\hat{d}_{\mathcal{K}}^2(X_5, X_6)$, $\hat{d}_{\mathcal{K}}^2(Y_5, Y_6)$, and $\hat{d}_{\mathcal{K}}^2((X_5, Y_5), (X_6, Y_6))$ might still be very close to zero, indicating high similarity at the local neighborhood level.

2. At a higher level, e.g. $l = 5$: As we increase the level $l$, the aggregated node features and labels capture a more global perspective of the graph structure. However, since the graphs have almost identical node features and labels in most parts, the MMD might still not be able to fully capture the differences due to the few differing edges.

These counter-examples demonstrate that the behavior of MMD between graphs can be influenced by the level $l$ of aggregation and the choice of kernel bandwidth. To alleviate the problem in the first counter-example, it is better to choose the kernel family $\mathcal{K}$ with a large set of bandwidths. As for the paradox within the second counter-example, there is no "one-size-fits-all" solution, because in some cases (e.g. unsupervised tasks), higher levels of aggregation might provide more informative comparisons, while in others (e.g. supervised tasks), local neighborhood information could be sufficient. It is worth noting that an increase in level $l$ may hurt the performance of the MMD graph kernel, since aggregation power would lost when the level $l$ exceeds the graph size itself.

## C  ALGORITHM OF MMD-GK

---
**Algorithm 2:** MMD-Graph Kernel

---
**Input**  : Graphs $G_1$ and $G_2$, number of level $L$, kernel family $\mathcal{K}$ with parameters $\{h_i\}_{i=1}^{|\mathcal{K}|}$, gamma $\gamma$
**Output:** Estimated similarity between $G_1$ and $G_2$
1  $Z_1^{(0)} \leftarrow (X_1, Y_1)$ or $Z_1^{(0)} \leftarrow X_1$ or $Z_1^{(0)} \leftarrow Y_1$; $Z_2^{(0)} \leftarrow (X_2, Y_2)$ or $Z_2^{(0)} \leftarrow X_2$ or $Z_2^{(0)} \leftarrow Y_2$
2  // Embedding Propgation
3  **for** $l = 1$ *to* $L$ **do**
4  $\quad\Big|\quad Z_1^{(l)} \leftarrow U_1 Z_1^{(l-1)}$ and $Z_2^{(l)} \leftarrow U_2 Z_2^{(l-1)}$
5  **end for**
6  // Kernel Mapping
7  $\hat{d}_{\mathcal{K}}^2(Z_1^{(L)}, Z_2^{(L)}) \leftarrow$ Eq. 6 with $Z_1^{(L)}, Z_2^{(L)}, \mathcal{K}$ and $\{h_i\}_{i=1}^{|\mathcal{K}|}$
8  // Distance-to-Similarity
9  $s_L(G_1, G_2) \leftarrow e^{-\gamma \cdot \hat{d}_{\mathcal{K}}^2(Z_1^{(l)}, Z_2^{(l)})}$
10  **return** $s_L(G_1, G_2)$

---

### C.1  ROBUSTNESS ANALYSIS

The bound in Theorem 1 provides insights into how the perturbed distance metric $\hat{d}_{\mathcal{K}}^2(\tilde{G}_1^{(l)}, \tilde{G}_2^{(l)})$ deviates from the original distance metric $\hat{d}_{\mathcal{K}}^2(G_1^{(l)}, G_2^{(l)})$. Specifically:

1) the deviation in MMD-GK grows bi-quadratically with $\|\Delta_Z\|_F$;

2) the deviation would be affected by $\|\Delta_A\|_2$ growing exponentially with power of $l$;

3) when $\Delta_A$ occurs, it also amplifies the perturbation $\|\Delta_Z\|_F$

4) $\Delta_{G_1}$ and $\Delta_{G_2}$ take an interaction effect in contaminating MMD-GK.

5) the deviation in MMD-GK is the most significant when $n = \frac{\epsilon^3}{8(\Delta_{G_1} + \Delta_{G_2})^3}$.

Reason: we take the derivative of $\bar{\Delta}$ w.r.t $n$ and observe the critical point.

$$\frac{\partial}{\partial n}\left(n(\Delta_{G_1} + \Delta_{G_2})^2 + \frac{\epsilon}{\sqrt{n}}(\Delta_{G_1} + \Delta_{G_2})\right) = (\Delta_{G_1} + \Delta_{G_2})^2 - \frac{\epsilon}{2\sqrt{n^3}}(\Delta_{G_1} + \Delta_{G_2}) \quad (12)$$

As $n \leq \frac{\epsilon^3}{8(\Delta_{G_1} + \Delta_{G_2})^3}$, the increase in $n$ leads to a larger $\bar{\Delta}$, which implies that the error due to perturbations becomes larger. And if $n \geq \frac{\epsilon^3}{8(\Delta_{G_1} + \Delta_{G_2})^3}$, the error due to perturbations becomes less significant when $n$ increases.

6) a larger $h$ implies a smoother kernel that may be more robust to perturbations.

# D    PROOF FOR THEOREMS

## D.1    PROOF OF THEOREM 1

*Proof.* For perturbations to node label $\tilde{Y} = Y + \Delta_Y$ and node attributes $\tilde{X} = X + \Delta_X$, we can define the perturbation on node features as $\tilde{Z} = Z + \Delta_Z$, where $\Delta_Z = (\Delta_X, \Delta_Y)$. When we introduce a perturbation $\Delta_A$ in $A$, it will also occurs in $D, U$, and ultimately takes effect on $\tilde{Z}^{(l)}$ as

$$\tilde{Z}^{(l)} = \underbrace{U' \cdots U'}_{l \text{ times}}(Z + \Delta_Z) = (U')^l Z + (U')^l \Delta_Z, \quad (13)$$

where $U' = \tilde{D}'^{-\frac{1}{2}}(\tilde{A} + \Delta_A)\tilde{D}'^{-\frac{1}{2}}$ and the perturbation $\tilde{D}' = \text{diag}(1^\top(\tilde{A} + \Delta_A))$. To simplify the further analysis, we denote $\tilde{Z}^{(l)} = U^l Z + \Delta_{Z^{(l)}}$ where the deviation can be expressed as

$$\Delta_{Z^{(l)}} = \left((U')^l - U^l\right)Z + (U')^l \Delta_Z.$$

For this deviation, we have the following lemma (proved in Section E.1).

**Lemma 1.** *Suppose the minimum node degrees of $G_1, G_2$ are both $\alpha$, $\|A_i\|_2 \leq \beta_A$, $\|X_i\|_2 \leq \beta_X$, $\|Y_i\|_2 \leq \beta_Y$, and $\|Z_i\|_F \leq \eta$, $i = 1, 2$. Denote the effects of structural perturbation as $\kappa = \min(1^\top \Delta_{A_i})$ and $\Delta_{D_i} = \text{diag}(1^\top(\tilde{A}_i + \Delta_{A_i}))^{\frac{1}{2}} \text{diag}(1^\top A_i)^{-\frac{1}{2}} - I$ for $i = 1, 2$. Then the following inequality holds:*

$$\|\Delta_{Z^{(l)}}\|_F \leq 2\eta \left(\frac{4\beta_A \|\Delta_D\|_2 + \|\Delta_A\|_2}{1 + \alpha}\right)^l + \left(\frac{2\beta_A + \|\Delta_A\|_2}{1 + \alpha + \kappa}\right)^l \|\Delta_Z\|_F. \quad (14)$$

Given a kernel family $\mathcal{K}$, the estimated generalized MMD between the perturbed $\tilde{G}_1^l$ and $\tilde{G}_2^l$ is $\hat{d}_{\mathcal{K}}^2(\tilde{G}_1^{(l)}, \tilde{G}_2^{(l)}) = \sup_{k \in \mathcal{K}} \hat{d}_k(\tilde{G}_1^{(l)}, \tilde{G}_2^{(l)})$, where

$$\hat{d}_k(\tilde{G}_1^{(l)}, \tilde{G}_2^{(l)}) = \frac{1}{n_1^2} \sum_{i,i'=1}^{n_1} k(\tilde{Z}_{i,1}^{(l)}, \tilde{Z}_{i',1}^{(l)}) + \frac{1}{n_2^2} \sum_{i,i'=1}^{n_2} k(\tilde{Z}_{j,2}^{(l)}, \tilde{Z}_{j',2}^{(l)}) - \frac{2}{n_1 n_2} \sum_{i,j=1}^{n_1,n_2} k(\tilde{Z}_{i,1}^{(l)}, \tilde{Z}_{j,2}^{(l)}). \quad (15)$$

For the original graphs $G_1^{(l)}$ and $G_2^{(l)}$, the distance can be expressed by replacing $\tilde{Z}$ with $Z$ in the equation above. Thus, we have the following inequality:

$$\begin{aligned}
\hat{d}_{\mathcal{K}}^2(\tilde{G}_1^{(l)}, \tilde{G}_2^{(l)}) - \hat{d}_{\mathcal{K}}^2(G_1^{(l)}, G_2^{(l)}) &= \sup_{k \in \mathcal{K}} \hat{d}_k(\tilde{G}_1^{(l)}, \tilde{G}_2^{(l)}) - \sup_{k \in \mathcal{K}} \hat{d}_k(G_1^{(l)}, G_2^{(l)}) \\
&\leq \hat{d}_{k^*}(\tilde{G}_1^{(l)}, \tilde{G}_2^{(l)}) - \hat{d}_{k^*}(G_1^{(l)}, G_2^{(l)}) \triangleq \bar{\Delta}_d,
\end{aligned} \quad (16)$$

where $k^* = \arg\max_{k \in \mathcal{K}} \hat{d}_k(\tilde{G}_1^{(l)}, \tilde{G}_2^{(l)})$. Since the difference primarily comes from the perturbation in kernel values, we can expand the expression as

$$
\begin{aligned}
\bar{\Delta}_d &= \frac{1}{n_1^2} \sum_{i,i'=1}^{n_1} \left( k^*(Z_{i,1}^{(l)}, Z_{i',1}^{(l)}) + \Delta k_{Z_{(1)}^{(l)}}^*(i,i') \right) + \frac{1}{n_2^2} \sum_{i,i'=1}^{n_2} \left( k^*(Z_{j,2}^{(l)}, Z_{j',2}^{(l)}) + \Delta k_{Z_{(2)}^{(l)}}^*(j,j) \right) \\
&\quad - \frac{2}{n_1 n_2} \sum_{i,j=1}^{n_1,n_2} \left( k^*(Z_{i,1}^{(l)}, Z_{j,2}^{(l)}) + \Delta k_{Z_{(1,2)}^{(l)}}^*(i,j') \right) \\
&= \frac{1}{n_1^2} \sum_{i,i'=1}^{n_1} \Delta k_{Z_{(1)}^{(l)}}^*(i,i') + \frac{1}{n_2^2} \sum_{j,j'=1}^{n_2} \Delta k_{Z_{(2)}^{(l)}}^*(j,j') - \frac{2}{n_1 n_2} \sum_{i,j=1}^{n_1,n_2} \Delta k_{Z_{(1,2)}^{(l)}}^*(i,j)
\end{aligned}
\tag{17}
$$

where $\Delta k_{Z_{(\cdot)}^{(l)}}^*(i,i') = k^*(\tilde{Z}_{i,\cdot}^{(l)}, \tilde{Z}_{i',\cdot}^{(l)}) - k^*(Z_{i,\cdot}^{(l)}, Z_{i',\cdot}^{(l)})$ and $\Delta k_{Z_{(1,2)}^{(l)}}^*(i,j) = k^*(\tilde{Z}_{i,1}^{(l)}, \tilde{Z}_{j,2}^{(l)}) - k^*(Z_{i,1}^{(l)}, Z_{j,2}^{(l)})$.

Without loss of generality, we take the Gaussian kernel as an example to further discuss the bound. The bandwidth of $k^*$ is denoted as $h$.

Next, we examine the basic components within the Gaussian kernel as Lemma 2 shows.

**Lemma 2.** *Suppose $\forall G, \|A\|_2 \le \beta_A, \|X\|_2 \le \beta_X, \|Y\|_2 \le \beta_Y$. Then all pairwise distances between l-level nodes of one graph are bounded as*

$$
\|z_{i,\cdot}^{(l)} - z_{i',\cdot}^{(l)}\| \le 2(1+\alpha)^{-l}(1+\beta_A)^l(\beta_X + \beta_Y).
$$

*And all pairwise distances between l-level nodes of two graphs are bounded as*

$$
\|z_{i,1}^{(l)} - z_{j,2}^{(l)}\| \le 2(1+\alpha)^{-l}(1+\beta_A)^l(\beta_X + \beta_Y).
$$

Given this and Lemma 4 in later context, both intra- and inter- exponential terms range between

$$
\exp(-\|z_{i,j}^{(l)} - z_{i',j}^{(l)}\|^2/h) \le 1 + \frac{8}{h^2}(1+\alpha)^{-4l}(1+\beta_A)^{4l}(\beta_X + \beta_Y)^4
$$

since $\|\Delta_{z_{i,\cdot}^{(l)}} - \Delta_{z_{i',\cdot}^{(l)}}\| \ge 0$ for all nodes.

Now, let's delve into the deviation in each kernel pairs as below.

**Lemma 3.** *Denote $\Sigma_{j,i,i',l} = \|\Delta_{z_{i,j}^{(l)}} - \Delta_{z_{i',j}^{(l)}}\|^2 - 2\|z_{i,j}^l - z_{i',j}^l\| \|\Delta_{z_{i,j}^{(l)}} - \Delta_{z_{i',j}^{(l)}}\|$. The following inequalities hold for an arbitrary Gaussian kernel $k$ with its bandwidth $h_k$:*

*(a)* $\Delta k_{Z_{(j)}^{(l)}}(i,i') \le \left( -\frac{\Sigma_{j,i,i',l}}{h_k} + \frac{\Sigma_{j,i,i',l}^2}{4h_k^2} \right) \exp\left( -\frac{\|z_{i,j}^{(l)} - z_{i',j}^{(l)}\|^2}{h_k} \right), \quad j = 1, 2.$

*(b)* $\Delta k_{Z_{(1,2)}^{(l)}}(i,j) \ge \left( \left( 1 - \frac{1}{h_k}\|\Delta_{z_{i,1}^{(l)}} + \Delta_{z_{j,2}^{(l)}}\|^2 \right) \left( 1 - \frac{2}{h_k}\|z_{i,1}^{(l)} - z_{j,2}^{(l)}\| \|\Delta_{z_{i,1}^{(l)}} + \Delta_{z_{j,2}^{(l)}}\| \right) - 1 \right) e^{-\frac{\|z_{i,1}^{(l)} - z_{j,2}^{(l)}\|^2}{h_k}}.$

Note that the bound for $\Sigma$ and its squared term is

$$
\begin{cases}
\Sigma_{\cdot,i,i',l} \ge -4(1+\alpha)^{-l}(1+\beta_A)^l(\beta_X + \beta_Y)\|\Delta_{z_{i,\cdot}^{(l)}} - \Delta_{z_{i',\cdot}^{(l)}}\| \\
\Sigma_{\cdot,i,i',l}^2 \le \|\Delta_{z_{i,\cdot}^{(l)}} - \Delta_{z_{i',\cdot}^{(l)}}\|^4 + 16(1+\alpha)^{-2l}(1+\beta_A)^{2l}(\beta_X + \beta_Y)^2\|\Delta_{z_{i,\cdot}^{(l)}} - \Delta_{z_{i',\cdot}^{(l)}}\|^2.
\end{cases}
$$

Consequently, the inequalities in Lemma 3 can be expressed as

$$
\begin{aligned}
\Delta k^*_{Z^{(l)}_{(j)}}(i, i') \leq & \left(1 + \frac{8}{h^2}(1+\alpha)^{-4l}(1+\beta_A)^{4l}(\beta_X + \beta_Y)^4\right) \cdot \\
& \left(\frac{4}{h}(1+\alpha)^{-l}(1+\beta_A)^l(\beta_X+\beta_Y)\|\Delta_{z^{(l)}_{i,\cdot}} - \Delta_{z^{(l)}_{i',\cdot}}\| + \frac{1}{4h^2}\|\Delta_{z^{(l)}_{i,\cdot}} - \Delta_{z^{(l)}_{i',\cdot}}\|^4 \right. \\
& \left. + \frac{4}{h^2}(1+\alpha)^{-2l}(1+\beta_A)^{2l}(\beta_X+\beta_Y)^2\|\Delta_{z^{(l)}_{i,\cdot}} - \Delta_{z^{(l)}_{i',\cdot}}\|^2\right)
\end{aligned}
\tag{18}
$$

and

$$
\begin{aligned}
\Delta k^*_{Z^{(l)}_{(1,2)}}(i, j) \geq & -\left(1 + \frac{8}{h^2}(1+\alpha)^{-4l}(1+\beta_A)^{4l}(\beta_X + \beta_Y)^4\right) \cdot \left(\frac{1}{h}\|\Delta_{z^{(l)}_{i,1}} + \Delta_{z^{(l)}_{j,2}}\|^2 \right. \\
& \left. + 4(1+\alpha)^{-l}(1+\beta_A)^l(\beta_X+\beta_Y)\|\Delta_{z^{(l)}_{i,1}} + \Delta_{z^{(l)}_{j,2}}\|\right).
\end{aligned}
\tag{19}
$$

Denote $\epsilon = 2(1+\alpha)^{-l}(1+\beta_A)^l(\beta_X + \beta_Y)$. We can summarize the inequalities as

$$
\begin{cases}
\Delta k^*_{Z^{(l)}_{(j)}}(i, i') \leq \left(1 + \frac{\epsilon^4}{2h^2}\right)\left(\frac{2}{h}\epsilon\|\Delta_{z^{(l)}_{i,\cdot}} - \Delta_{z^{(l)}_{i',\cdot}}\| + \frac{1}{h^2}\left(\frac{1}{2}\|\Delta_{z^{(l)}_{i,\cdot}} - \Delta_{z^{(l)}_{i',\cdot}}\|^2 + \epsilon\right)^2 - \frac{\epsilon^2}{h}\right) \\
\Delta k^*_{Z^{(l)}_{(1,2)}}(i, j) \geq -\left(1 + \frac{\epsilon^4}{2h^2}\right)\frac{1}{h}\left(\left(\|\Delta_{z^{(l)}_{i,1}} + \Delta_{z^{(l)}_{j,2}}\| + \epsilon\right)^2 - \epsilon^2\right).
\end{cases}
\tag{20}
$$

By substituting Eq. 20 into Eq. 17, the deviation is bounded by a function of $\Delta_z$:

$$
\begin{aligned}
\bar{\Delta}_d \leq & \left(\frac{1}{n_1^2 h^2} + \frac{\epsilon^4}{2n_1^2 h^4}\right)\sum_{i,i'=1}^{n_1}\left[\left(\frac{1}{2}\|\Delta_{z^{(l)}_{i,\cdot}} - \Delta_{z^{(l)}_{i',\cdot}}\|^2 + \epsilon\right)^2 + 2h\epsilon\|\Delta_{z^{(l)}_{i,\cdot}} - \Delta_{z^{(l)}_{i',\cdot}}\| - \epsilon^2\right] \\
& + \left(\frac{1}{n_2^2 h^2} + \frac{\epsilon^4}{2n_2^2 h^4}\right)\sum_{j,j'=1}^{n_2}\left[\left(\frac{1}{2}\|\Delta_{z^{(l)}_{j,\cdot}} - \Delta_{z^{(l)}_{j',\cdot}}\|^2 + \epsilon\right)^2 + 2h\epsilon\|\Delta_{z^{(l)}_{j,\cdot}} - \Delta_{z^{(l)}_{j',\cdot}}\| - \epsilon^2\right] \\
& + \left(\frac{2}{n_1 n_2 h} + \frac{\epsilon^4}{n_1 n_2 h^3}\right)\sum_{i,j=1}^{n_1,n_2}\left[\left(\|\Delta_{z^{(l)}_{i,1}} + \Delta_{z^{(l)}_{j,2}}\| + \epsilon\right)^2 - \epsilon^2\right].
\end{aligned}
\tag{21}
$$

Based on Equation 35 and $\|\Delta_{Z^{(l)}}\|_{2,1} \leq \sqrt{n}\|\Delta_{Z^{(l)}}\|_F$, we can obtain the following inequalities which apply to every individual graph $G_{\cdot}$ and pair $(G_1, G_2)$.

$$
\sum_{i,i'}^{n_\cdot}\|\Delta_{z^{(l)}_{i,\cdot}} - \Delta_{z^{(l)}_{i',\cdot}}\| \leq \sum_{i,i'}^{n_\cdot}\left(\|\Delta_{z^{(l)}_{i,\cdot}}\| + \|\Delta_{z^{(l)}_{i',\cdot}}\|\right) = 2n_\cdot\|\Delta_{Z^{(l)}_\cdot}\|_{2,1} \leq 2n_\cdot^{\frac{3}{2}}\Delta_{G_\cdot}
\tag{22a}
$$

$$
\sum_{i,i'}^{n_\cdot}\|\Delta_{z^{(l)}_{i,\cdot}} - \Delta_{z^{(l)}_{i',\cdot}}\|^2 \leq \sum_{i,i'}^{n_\cdot}\left(\|\Delta_{z^{(l)}_{i,\cdot}}\| + \|\Delta_{z^{(l)}_{i',\cdot}}\|\right)^2 = 2n_\cdot\|\Delta_{Z^{(l)}_\cdot}\|_F^2 + 2\|\Delta_{Z^{(l)}_\cdot}\|_{2,1}^2 \leq 4n_\cdot(\Delta_{G_\cdot})^2
\tag{22b}
$$

$$
\begin{aligned}
\sum_{i,j}^{n_1,n_2}\|\Delta_{z^{(l)}_{i,1}} + \Delta_{z^{(l)}_{j,2}}\| & \leq n_2\sum_i^{n_1}\|\Delta_{z^{(l)}_{i,1}}\| + n_1\sum_j^{n_2}\|\Delta_{z^{(l)}_{j,2}}\| = n_1\|\Delta_{Z^{(l)}_2}\|_{2,1} + n_2\|\Delta_{Z^{(l)}_1}\|_{2,1} \\
& \leq n_1\sqrt{n_2}\Delta_{G_2} + n_2\sqrt{n_1}\Delta_{G_1}
\end{aligned}
\tag{22c}
$$

where $\Delta_{G.} = 2\eta \left( \frac{4\beta_A \|\Delta_{D.}\|_2 + \|\Delta_{A.}\|_2}{1+\alpha} \right)^l + \left( \frac{2\beta_A + \|\Delta_{A.}\|_2}{1+\alpha+\kappa} \right)^l \|\Delta_{Z.}\|_F$. To sum up, the deviation in MMD-GK is bounded as

$$\hat{d}_{\mathcal{K}}^2(\tilde{G}_1^{(l)}, \tilde{G}_2^{(l)}) - \hat{d}_{\mathcal{K}}^2(G_1^{(l)}, G_2^{(l)}) \leq \left( \frac{1}{h^2} + \frac{\epsilon^4}{2h^4} \right) \left[ \left( 2\Delta_{G_1}^2 + \frac{\epsilon}{n_1} \right)^2 + \left( 2\Delta_{G_2}^2 + \frac{\epsilon}{n_2} \right)^2 - \frac{\epsilon^2}{n_1^2} - \frac{\epsilon^2}{n_2^2} \right.$$
$$\left. + 4h\epsilon \left( \frac{\Delta_{G_1}}{\sqrt{n_1}} + \frac{\Delta_{G_2}}{\sqrt{n_2}} \right) \right]$$
$$+ \left( \frac{2}{h} + \frac{\epsilon^4}{h^3} \right) \left[ \left( \sqrt{n_1}\Delta_{G_2} + \sqrt{n_2}\Delta_{G_1} + \frac{\epsilon}{\sqrt{n_1 n_2}} \right)^2 - \frac{\epsilon^2}{n_1 n_2} \right]$$
$$\tag{23}$$

If $n_1 = n_2 = n$, we can simplify it as

$$\hat{d}_{\mathcal{K}}^2(\tilde{G}_1^{(l)}, \tilde{G}_2^{(l)}) - \hat{d}_{\mathcal{K}}^2(G_1^{(l)}, G_2^{(l)}) \leq \left( \frac{2}{h} + \frac{\epsilon^4}{h^3} \right) \left( \frac{\left( 2\Delta_{G_1}^2 + \epsilon/n \right)^2 + \left( 2\Delta_{G_2}^2 + \epsilon/n \right)^2}{2h} - \frac{\epsilon^2}{n^2 h} \right.$$
$$\left. + \frac{2\epsilon}{\sqrt{n}} (\Delta_{G_1} + \Delta_{G_2}) + \left( \sqrt{n}(\Delta_{G_1} + \Delta_{G_2}) + \frac{\epsilon}{n} \right)^2 - \frac{\epsilon^2}{n^2} \right) \tag{24}$$

After further simplification, we eventually have

$$\hat{d}_{\mathcal{K}}^2(\tilde{G}_1^{(l)}, \tilde{G}_2^{(l)}) - \hat{d}_{\mathcal{K}}^2(G_1^{(l)}, G_2^{(l)}) \leq \left( \frac{4}{h^2} + \frac{2\epsilon^4}{h^4} \right) \left( 2\Delta_{G_1}^4 + 2\Delta_{G_2}^4 + n(\Delta_{G_1} + \Delta_{G_2})^2 + \frac{\epsilon}{\sqrt{n}} (\Delta_{G_1} + \Delta_{G_2}) \right) \tag{25}$$

$$\square$$

### D.2 PROOF OF THEOREM 2

*Proof.* For Deep MMD-GK,

$$Z^{(l)} = \underbrace{\sigma(U \cdots \sigma(U}_{l \text{ times}} Z \underbrace{W^{(1)}) \cdots W^{(l)}}_{l \text{ times}}) \tag{26}$$

Similar to proof of Theorem D.1, a perturbation $\Delta_A$ in $A$, will ultimately takes effect on $\tilde{Z}^{(l)}$ as

$$\tilde{Z}^{(l)} = \underbrace{\sigma(U' \cdots \sigma(U'}_{l \text{ times}} (Z + \Delta_Z) \underbrace{W^{(1)}) \cdots W^{(l)}}_{l \text{ times}}) \tag{27}$$

Denote $\tilde{Z}^{(l)} = Z^{(l)} + \Delta_{Z^{(l)}}$ where the deviation can be expressed as $\Delta_{Z^{(l)}} = \tilde{Z}^{(l)} - Z^{(l)}$. Let $\alpha$ be the minimum node degree of $G$, we have $\|\tilde{D}^{-\frac{1}{2}}\|_2 \leq (1+\alpha)^{-1/2}$. Assume that the norm and $\|\tilde{A}\|_2 \leq (\beta_A + 1)$, so that $\|U\|_2 \leq (1+\alpha)^{-1}\beta_A$, $\|U'\|_2 \leq (1+\alpha+\kappa)^{-1}(\beta_A + \|\Delta_A\|_2)$ where $\kappa = \min(1^\top \Delta_A)$, and $\|U' - U\|_2^2 \leq (1+\alpha+\kappa)^{-2} \left( (\beta_A + \|\Delta_A\|_2)^2 - \beta_A^2 \right)$. Suppose $\|Z\|_F \leq \eta$, $\|W^{(l)}\|_2 \leq \beta_{W^{(l)}}$, and the activation

function $\sigma$ is $\rho$-Lipschitz continuous. Based on Eq. 26, $\|Z^{(l-1)}\|_F \leq \rho^{l-1}(\frac{\beta_A}{1+\alpha})^{l-1}\prod_{i=1}^{l-1}\beta_{W^{(i)}}\|Z\|_F$, then we derive that

$$
\begin{aligned}
\|\Delta_{Z^{(l)}}\|_F &\leq \rho\left\|U'\tilde{Z}^{(l-1)} - UZ^{(l-1)}\right\|_F\|W^{(l)}\|_2 \\
&\leq \rho\left(\|U'\|_2\|\Delta_{Z^{(l-1)}}\|_F + \|U'-U\|_2\|Z^{(l-1)}\|_F\right)\|W^{(l)}\|_2 \\
&\leq \rho\beta_{W^{(l)}}(1+\alpha+\kappa)^{-1}\left((\beta_A+\|\Delta_A\|_2)\|\Delta_{Z^{(l-1)}}\|_F + \sqrt{(\beta_A+\|\Delta_A\|_2)^2-\beta_A^2}\|Z^{(l-1)}\|_F\right) \\
&\leq \rho^l\prod_{i=1}^{l-1}\beta_{W^{(i)}}(1+\alpha+\kappa)^{-l}(\beta_A+\|\Delta_A\|_2)^l(\|\Delta_Z\|_F) \\
&\quad + \sum_{j=0}^{l}\rho^{l-j}\prod_{i=1+j}^{l}\beta_{W^{(i)}}\left(\frac{\sqrt{(\beta_A+\|\Delta_A\|_2)^2-\beta_A^2}}{1+\alpha+\kappa}\right)^{l-j}\rho^j\left(\frac{\beta_A}{1+\alpha+\kappa}\right)^j\prod_{i=1}^{j}\beta_{W^{(i)}}\|Z\|_F \\
&= \left(\frac{\rho}{1+\alpha+\kappa}\right)^l\prod_{i=1}^{l}\beta_{W^{(i)}}\left((\beta_A+\|\Delta_A\|_2)^l\|\Delta_Z\|_F + \eta\sum_{j=0}^{l-1}\beta_A^j\left(\sqrt{(\beta_A+\|\Delta_A\|_2)^2-\beta_A^2}\right)^{l-j}\right)
\end{aligned}
\tag{28}
$$

Other things being equal to that of Theorem 1, the deviation in Deep MMD-GK satisfies the following inequality if $n_1 = n_2 = n$,

$$
\hat{d}_{\mathcal{K}}^2(\tilde{G}_1^{(l)}, \tilde{G}_2^{(l)}) - \hat{d}_{\mathcal{K}}^2(G_1^{(l)}, G_2^{(l)}) \leq \left(\frac{4}{h^2} + \frac{2\epsilon^4}{h^4}\right)\left(2\Delta_{G_1}^4 + 2\Delta_{G_2}^4 + n(\Delta_{G_1}+\Delta_{G_2})^2 + \frac{\epsilon}{\sqrt{n}}(\Delta_{G_1}+\Delta_{G_2})\right)
\tag{29}
$$

where $\Delta_{G.} = \left(\frac{\rho}{1+\alpha+\kappa}\right)^l\prod_{i=1}^{l}\beta_{W^{(i)}}\left((\beta_A+\|\Delta_{A.}\|_F)^l\|\Delta_Z\|_F + \eta\sum_{j=0}^{l-1}\beta_A^j\left(\sqrt{(\beta_A+\|\Delta_{A.}\|_2)^2-\beta_A^2}\right)^{l-j}\right)$.
$\square$

The implications are similar to Theorem 1 that

1) the deviation in Deep MMD-GK grows bi-quadratically with $\|\Delta_Z\|_F$;
2) the deviation would be affected by $\|\Delta_A\|_2$ growing exponentially with power of $l$;
3) when $\Delta_A$ occurs, it also amplifies the perturbation $\|\Delta_Z\|_F$
4) $\Delta_{G_1}$ and $\Delta_{G_2}$ take an interaction effect in contaminating Deep MMD-GK.
5) the deviation in Deep MMD-GK is the most significant when $n = \frac{\epsilon^3}{8(\Delta_{G_1}+\Delta_{G_2})^3}$.
6) a larger $h$ implies a smoother kernel that may be more robust to perturbations.

# E  PROOF FOR LEMMAS

## E.1  PROOF FOR LEMMA 1

Let $\alpha$ be the minimum node degree of $G$, we have $\|\tilde{D}^{-\frac{1}{2}}\|_2 \leq (1+\alpha)^{-1/2}$. Suppose $\|A\|_2 \leq \beta_A$, then $\|\tilde{A}\|_2 \leq (1+\beta_A)$. We can obtain

$$
\|U\|_2 \leq \|\tilde{D}^{-\frac{1}{2}}\|_2\|\tilde{A}\|_2\|\tilde{D}^{-\frac{1}{2}}\|_2 \leq (1+\alpha)^{-1}(1+\beta_A).
\tag{30}
$$

Similarly, we have

$$\|U'\|_2 \le \|(\tilde{D}')^{-\frac{1}{2}}\|_2^2 \Big(\|\tilde{A}\|_2 + \|\Delta_A\|_2\Big) \le (1+\alpha+\kappa)^{-1}(1+\beta_A+\|\Delta_A\|_2) \tag{31}$$

where $\kappa = \min(1^\top \Delta_A)$. It follows that

$$
\begin{aligned}
\|U'-U\|_2 &= \|\tilde{D}'^{-\frac{1}{2}}(\tilde{A}+\Delta_A)\tilde{D}'^{-\frac{1}{2}} - \tilde{D}^{-\frac{1}{2}}\tilde{A}\tilde{D}^{-\frac{1}{2}}\|_2 \\
&= \|\tilde{D}'^{-\frac{1}{2}}\tilde{A}\tilde{D}'^{-\frac{1}{2}} - \tilde{D}'^{-\frac{1}{2}}\tilde{A}\tilde{D}^{-\frac{1}{2}} + \tilde{D}'^{-\frac{1}{2}}\tilde{A}\tilde{D}^{-\frac{1}{2}} - \tilde{D}^{-\frac{1}{2}}\tilde{A}\tilde{D}^{-\frac{1}{2}} + \tilde{D}'^{-\frac{1}{2}}\Delta_A\tilde{D}'^{-\frac{1}{2}}\|_2 \\
&\le \|\tilde{D}'^{-\frac{1}{2}}\tilde{A}(\tilde{D}'^{-\frac{1}{2}} - \tilde{D}^{-\frac{1}{2}})\|_2 + \|(\tilde{D}'^{-\frac{1}{2}} - \tilde{D}^{-\frac{1}{2}})\tilde{A}\tilde{D}^{-\frac{1}{2}}\|_2 + \|\tilde{D}'^{-\frac{1}{2}}\Delta_A\tilde{D}'^{-\frac{1}{2}}\|_2 \\
&\le (1+\beta_A)((1+\alpha+\kappa)^{-1/2} + (1+\alpha)^{-1/2})\|\tilde{D}'^{-\frac{1}{2}} - \tilde{D}^{-\frac{1}{2}}\|_2 + (1+\alpha+\kappa)^{-1}\|\Delta_A\|_2 \\
&\le 2(1+\beta_A)(1+\alpha)^{-1/2}\|\tilde{D}'^{-\frac{1}{2}} - \tilde{D}^{-\frac{1}{2}}\|_2 + (1+\alpha+\kappa)^{-1}\|\Delta_A\|_2 \\
&\le 2(1+\beta_A)(1+\alpha)^{-1/2}(1+\alpha+\kappa)^{-1/2}\|I - \tilde{D}'^{\frac{1}{2}}\tilde{D}^{-\frac{1}{2}}\|_2 + (1+\alpha+\kappa)^{-1}\|\Delta_A\|_2 \\
&\triangleq 2(1+\beta_A)(1+\alpha)^{-1/2}(1+\alpha+\kappa)^{-1/2}\|\Delta_D\|_2 + (1+\alpha+\kappa)^{-1}\|\Delta_A\|_2 \\
&\le 2(1+\beta_A)(1+\alpha)^{-1}\|\Delta_D\|_2 + (1+\alpha+\kappa)^{-1}\|\Delta_A\|_2
\end{aligned}
\tag{32}
$$

where $\|\Delta_D\|_2 = \|I - \tilde{D}'^{\frac{1}{2}}\tilde{D}^{-\frac{1}{2}}\|_2$.

$$
\begin{aligned}
\|(U')^l - U^l\|_2 &\le \|U'\|_2\|(U')^{l-1} - U^{l-1}\|_2 + \|U'-U\|_2\|U^{l-1}\|_2 \\
&\le \Big(2(1+\beta_A)(1+\alpha)^{-1}\|\Delta_D\|_2 + (1+\alpha+\kappa)^{-1}\|\Delta_A\|_2\Big)\left(\frac{1+\beta_A+\|\Delta_A\|_2}{1+\alpha+\kappa}\right)^{l-1} \\
&\quad + \Big(2(1+\beta_A)(1+\alpha)^{-1}\|\Delta_D\|_2 + (1+\alpha+\kappa)^{-1}\|\Delta_A\|_2\Big)\sum_{i=1}^{l-1}\left(\frac{1+\beta_A}{1+\alpha}\right)^{l-i}\left(\frac{1+\beta_A+\|\Delta_A\|_2}{1+\alpha+\kappa}\right)^i \\
&\le \left(\frac{2(1+\beta_A)\|\Delta_D\|_2}{1+\alpha} + \frac{\|\Delta_A\|_2}{1+\alpha+\kappa}\right)\left(\left(\frac{1+\beta_A+\|\Delta_A\|_2}{1+\alpha+\kappa}\right)^{l-1} + \sum_{i=1}^{l-1}\left(\frac{1+\beta_A}{1+\alpha}\right)^{l-i}\left(\frac{1+\beta_A+\|\Delta_A\|_2}{1+\alpha+\kappa}\right)^i\right) \\
&\le 2\left(\frac{2(1+\beta_A)\|\Delta_D\|_2}{1+\alpha} + \frac{\|\Delta_A\|_2}{1+\alpha+\kappa}\right)\left(\frac{1+\beta_A}{1+\alpha} + \frac{1+\beta_A+\|\Delta_A\|_2}{1+\alpha+\kappa}\right)^{l-1}
\end{aligned}
\tag{33}
$$

The Frobenius norm of the deviation in $\Delta_{Z^{(l)}}$ is bounded as

$$
\begin{aligned}
\|\Delta_{Z^{(l)}}\|_F &\le \|\big((U')^l - U^l\big)Z\|_F + \|(U')^l\Delta_Z\|_F \\
&\le \|(U')^l - U^l\|_2\|Z\|_F + \|U'\|_2^l\|\Delta_Z\|_F
\end{aligned}
\tag{34}
$$

Given Eq. 31, 33, and assume $\|Z\|_F \leq \eta$, the bound could be expressed as

$$
\begin{aligned}
\|\Delta_{Z^{(l)}}\|_F &\leq 2\eta \left( \frac{2(1+\beta_A)\|\Delta_D\|_2}{1+\alpha} + \frac{\|\Delta_A\|_2}{1+\alpha+\kappa} \right) \left( \frac{1+\beta_A}{1+\alpha} + \frac{1+\beta_A+\|\Delta_A\|_2}{1+\alpha+\kappa} \right)^{l-1} + \left( \frac{1+\beta_A+\|\Delta_A\|_2}{1+\alpha+\kappa} \right)^l \|\Delta_Z\|_F \\
&\leq 2\eta \left( \frac{2(1+\beta_A)\|\Delta_D\|_2+\|\Delta_A\|_2}{1+\alpha} \right) \left( \frac{2(1+\beta_A)+\|\Delta_A\|_2}{1+\alpha} \right)^{l-1} + \left( \frac{1+\beta_A+\|\Delta_A\|_2}{1+\alpha+\kappa} \right)^l \|\Delta_Z\|_F \\
&\leq 2\eta \left( \frac{2(1+\beta_A)\|\Delta_D\|_2+\|\Delta_A\|_2}{1+\alpha} \right)^l + \left( \frac{1+\beta_A+\|\Delta_A\|_2}{1+\alpha+\kappa} \right)^l \|\Delta_Z\|_F \\
&\overset{i}{\leq} 2\eta \left( \frac{4\beta_A\|\Delta_D\|_2+\|\Delta_A\|_2}{1+\alpha} \right)^l + \left( \frac{2\beta_A+\|\Delta_A\|_2}{1+\alpha+\kappa} \right)^l \|\Delta_Z\|_F,
\end{aligned}
\tag{35}
$$

where inequality $i$ used the fact that $\beta_A \geq 1$.

### E.2 PROOF FOR LEMMA 3

**Lemma 4.** $e^x \leq 1 + x + \frac{x^2}{2!}$ when $x \leq 0$.

First, based on Lemma 4, we can bound $\Delta k_{Z_{(1)}^{(l)}}(i, i')$ as follows:

$$
\begin{aligned}
\Delta k_{Z_{(1)}^{(l)}}(i, i') &= \exp\left( -\frac{\|z_{i,1}^{(l)} + \Delta_{z_{i,1}^{(l)}} - (z_{i',1}^{(l)} + \Delta_{z_{i',1}^{(l)}})\|^2}{h_k} \right) - \exp\left( -\frac{\|z_{i,1}^{(l)} - z_{i',1}^{(l)}\|^2}{h_k} \right) \\
&\leq \exp\left( -\frac{\|z_{i,1}^{(l)} - z_{i',1}^{(l)}\|^2 + \|\Delta_{z_{i,1}^{(l)}} - \Delta_{z_{i',1}^{(l)}}\|^2 - 2\|z_{i,1}^{(l)} - z_{i',1}^{(l)}\|\|\Delta_{z_{i,1}^{(l)}} - \Delta_{z_{i',1}^{(l)}}\|}{h_k} \right) - \exp\left( -\frac{\|z_{i,1}^{(l)} - z_{i',1}^{(l)}\|^2}{h_k} \right) \\
&= \left[ \exp\left( -\frac{\|\Delta_{z_{i,1}^{(l)}} - \Delta_{z_{i',1}^{(l)}}\|^2 - 2\|z_{i,1}^{(l)} - z_{i',1}^{(l)}\|\|\Delta_{z_{i,1}^{(l)}} - \Delta_{z_{i',1}^{(l)}}\|}{h_k} \right) - 1 \right] \exp\left( -\frac{\|z_{i,1}^{(l)} - z_{i',1}^{(l)}\|^2}{h_k} \right) \\
&\leq \left( -\frac{\Sigma_{1,i,i',l}}{h_k} + \frac{\Sigma_{1,i,i',l}^2}{4h_k^2} \right) \exp\left( -\frac{\|z_{i,1}^{(l)} - z_{i',1}^{(l)}\|^2}{h_k} \right)
\end{aligned}
\tag{36}
$$

where $\Sigma_{1,i,i',l} = \|\Delta_{z_{i,1}^{(l)}} - \Delta_{z_{i',1}^{(l)}}\|^2 - 2\|z_{i,1}^l - z_{i',1}^l\|\|\Delta_{z_{i,1}^{(l)}} - \Delta_{z_{i',1}^{(l)}}\|$. The related deduction for $\Delta k_{Z_{(2)}^{(l)}}(j, j')$ is almost the same as (36) and hence is omitted.

For $\Delta k_{Z^{(l)}_{(1,2)}}(i,j)$, we have

$$\Delta k_{Z^{(l)}_{(1,2)}}(i,j) = \exp\left(-\frac{\|z^{(l)}_{i,1}+\Delta_{z^{(l)}_{i,1}}-z^{(l)}_{j,2}-\Delta_{z^{(l)}_{j,2}}\|^2}{h_k}\right) - \exp\left(-\frac{\|z^{(l)}_{i,1}-z^{(l)}_{j,2}\|^2}{h_k}\right)$$

$$\geq \exp\left(-\frac{\|z^{(l)}_{i,1}-z^{(l)}_{j,2}\|^2+\|\Delta_{z^{(l)}_{i,1}}+\Delta_{z^{(l)}_{j,2}}\|^2+2\|z^{(l)}_{i,1}-z^{(l)}_{j,2}\|\|\Delta_{z^{(l)}_{i,1}}+\Delta_{z^{(l)}_{j,2}}\|}{h_k}\right) - \exp\left(-\frac{\|z^{(l)}_{i,1}-z^{(l)}_{j,2}\|^2}{h_k}\right)$$

$$\overset{i}{\geq} \left(1-\frac{1}{h_k}\|\Delta_{z^{(l)}_{i,1}}+\Delta_{z^{(l)}_{j,2}}\|^2\right)\left(1-\frac{2}{h_k}\|z^{(l)}_{i,1}-z^{(l)}_{j,2}\|\|\Delta_{z^{(l)}_{i,1}}+\Delta_{z^{(l)}_{j,2}}\|\right)e^{-\frac{\|z^{(l)}_{i,1}-z^{(l)}_{j,2}\|^2}{h_k}} - e^{-\frac{\|z^{(l)}_{i,1}-z^{(l)}_{j,2}\|^2}{h_k}}$$

(37)

where inequality $i$ used the fact $e^x > 1 + x$.

### E.3 PROOF FOR LEMMA 4

*Proof.* We want to show $f(x) > 0$ for all $x$, where $f : (-\infty, 0] \to \mathbb{R}$ is the function defined by $f(x) = 1 + x + \frac{x^2}{2!} + \cdots + \frac{x^n}{n} - e^x$. Since $f(x) \to \infty$ as $x \to -\infty$, $f$ must attain an absolute minimum somewhere on the interval $(-\infty, 0]$.

1. If $f$ has an absolute minimum at 0, then for all $x$, $f(x) \geq f(0) = 1 - e^0 = 0$, so it is proved.

2. If $f$ has an absolute minimum at $y$ for some $y < 0$, then $f'(y) = 0$. But differentiating $f$,

$$f'(y) = 1 + y + \frac{y^2}{2!} + \cdots + \frac{y^{n-1}}{(n-1)!} - e^y = f(y) - \frac{y^n}{n!}.$$

Set $n = 2$. For all $x \leq 0$, $f(x) \geq f(y) = \frac{y^2}{4} + f'(y) = \frac{y^2}{4} > 0$. Then $f(x) = 1 + x + \frac{x^2}{4} - e^x > 0$, which means that

$$e^x \leq 1 + x + \frac{x^2}{4}$$

$\square$

### E.4 PROOF FOR LEMMA 2

*Proof.* Given that $\|X\|_2 \leq \beta_X$, $\|Y\|_2 \leq \beta_Y$, $\|\tilde{D}^{-1/2}_{i,:}\|_2 \leq (1+\alpha)^{-1/2}$, $\|\tilde{D}^{-\frac{1}{2}}\|_2 \leq (1+\alpha)^{-1/2}$, $\|\tilde{A}\|_2 \leq (1+\beta_A)$, and $\|U\|_2 \leq (1+\alpha)^{-1}(1+\beta_A)$ in Eq. 30, we then obtain the following upper bound step by step:

$$
\begin{aligned}
&\|z^{(l)}_{i,:} - z^{(l)}_{i',:}\| \\
=&\|U_{i,:}U^{l-1}Z - U_{i',:}U^{l-1}Z\| \\
\leq&\|U_{i,:} - U_{i',:}\|\|U^{l-1}\|_2\|Z\|_2 \\
\leq&\|\tilde{D}^{-1/2}_{i,:}\tilde{A}\tilde{D}^{-1/2} - \tilde{D}^{-1/2}_{i',:}\tilde{A}\tilde{D}^{-1/2}\|\|U\|^{l-1}_2\|Z\|_2 \\
\leq&\|\tilde{D}^{-1/2}_{i,:} - \tilde{D}^{-1/2}_{i',:}\|\|\tilde{A}\|_2\|\tilde{D}^{-1/2}\|_2\|U\|^{l-1}_2\|Z\|_2 \\
\leq&(\|\tilde{D}^{-1/2}_{i,:}\|+\|\tilde{D}^{-1/2}_{i',:}\|)(1+\beta_A)(1+\alpha)^{-1/2}(1+\alpha)^{-l+1}(1+\beta_A)^{l-1}\|Z\|_2 \\
\leq&2(1+\alpha)^{-1/2}(1+\beta_A)^l(1+\alpha)^{-l+1/2}(\|X\|_2+\|Y\|_2) \\
\leq&2(1+\alpha)^{-l}(1+\beta_A)^l(\beta_X+\beta_Y)
\end{aligned}
$$

(38)

Suppose $\|X_i\|_2 \leq \beta_X$, $\|Y_i\|_2 \leq \beta_Y$, $\|A_i\| \leq \beta_A$ and $\|\tilde{D}_i^{-\frac{1}{2}}\|_2 \leq (1+\alpha)^{-1/2} \ \forall i = 1, 2$, we have $\|U_i\| \leq (1+\alpha)^{-1}(1+\beta_A)$. Consequently, we then find the same upper bound for $\|z_{i,1}^{(l)} - z_{j,2}^{(l)}\|$:

$$
\begin{aligned}
&\|z_{i,1}^{(l)} - z_{j,2}^{(l)}\| \\
=&\|U_{1,i,:}U_1^{l-1}Z_1 - U_{2,j,:}U_2^{l-1}Z_2\| \\
\leq&\|U_{1,i,:}U_1^{l-1}Z_1\| + \|U_{2,j,:}U_2^{l-1}Z_2\| \\
\leq&\|U_{1,i,:}\|\|U_1^{l-1}\|_2\|Z_1\|_2 + \|U_{2,j,:}\|\|U_2^{l-1}\|_2\|Z_2\|_2 \\
\leq&\|\tilde{D}_{1,i,:}\|\|\tilde{A}_1\|\|\tilde{D}_1^{-1/2}\|\|U_1\|_2^{l-1}\|Z_1\|_2 + \|\tilde{D}_{2,j,:}\|\|\tilde{A}_2\|\|\tilde{D}_2^{-1/2}\|\|U_2\|_2^{l-1}\|Z_2\|_2 \\
\leq&2(1+\alpha)^{-1/2}(1+\beta_A)(1+\alpha)^{-1/2}(1+\alpha)^{-l+1}(1+\beta_A)^{l-1}(\beta_X + \beta_Y) \\
\leq&2(1+\alpha)^{-l}(1+\beta_A)^l(\beta_X + \beta_Y)
\end{aligned}
\tag{39}
$$

$\square$

## F    GENERALIZATION ERROR FOR SUPERVISED LOSS

For the convenience, we write $\mathcal{L}_{\text{SCL}} = l(W, r)$ to highlight its dependence on $W$ and $r = (Z_u, C_u, Z_v, C_v)$, a pair of graph $u$ and $v$ in the training data. Let $W_{\mathcal{D}} := \{W^{(i)}\}_{i=1}^L$ be the parameters learned from the training data $\mathcal{D}$ through Algorithm 1. By removing the $i$-th element in $\mathcal{D}$ with $N$ total pairs of graphs, we define $\mathcal{D}^{\setminus i} = \{r_1, \cdots, r_{i-1}, r_{i+1}, \cdots, r_N\}$. To get bounds on the generalization error, we consider that a stronger notion of stability will allow us to get tighter bounds, one of which is the *uniform stability* (Bousquet & Elisseeff, 2002; Feldman & Vondrak, 2019). The uniform stability of an algorithm determines its stability when one of the training data is removed. The goal is to find $\omega$ such that for any training set $\mathcal{D}$ and any pair $r_i$, the following inequality holds:

$$
\sup_{\{u,v\}} |l(W_{\mathcal{D}}, Z_u, Z_v) - l(W_{\mathcal{D}^{\setminus i}}, Z_u, Z_v)| \leq \omega \qquad \forall i \in \{1, \cdots, N\}
\tag{40}
$$

**Theorem 5.** *Given the parameters are known, and $d_{u,v}^2(W_{\mathcal{D}}) \leq \tau$ for any pairs of graphs $u, v$, Algorithm 1 with the supervised contrastive loss (Eq. 8) is guaranteed with a uniform stability parameters $\omega$ as below*

$$
\omega = 4\Psi + 2\gamma\tau + 2\log(1 + \Psi),
\tag{41}
$$

*where $\Psi = \gamma\left(\frac{2}{h^2} + \frac{\epsilon^4}{h^4}\right)\left(2\Delta_G^4 + 2n\Delta_G^2 + \frac{\epsilon}{\sqrt{n}}\Delta_G\right)$, and the deviation in graph via learned parameters is denoted as $\Delta_G = \eta\left(\frac{\rho\beta_A}{1+\alpha}\right)^l \sum_{j=0}^l\left(\prod_{t=1+j}^l \beta_{W_{\mathcal{D}^{\setminus i}}^{(t)}} \prod_{t=1}^j \|W_{\mathcal{D}^{\setminus i}}^{(j)} - W_{\mathcal{D}}^{(j)}\|_F\right)$.*

*Proof.* Given that $s_{uv} = \exp(-\gamma d_{uv}^2)$, we denote squared MMD $d_{uv}^2 := d_{uv}^2(W_{\mathcal{D}})$ as a function of the parameters learned from the training dataset $\mathcal{D}$.

$$
\begin{aligned}
|l(W_{\mathcal{D}}, Z_u, Z_v) - l(W_{\mathcal{D}^{\setminus i}}, Z_u, Z_v)| = &\Bigg| \gamma \cdot \left(d_{uv}^2(W_{\mathcal{D}^{\setminus i}}) - d_{uv}^2(W_{\mathcal{D}})\right) + \\
&\log \frac{\sum_k \mathbb{I}_{[C_u = C_k, u \neq k]} \exp(-\gamma d_{uk}^2(W_{\mathcal{D}^{\setminus i}})) + \alpha \sum_k \mathbb{I}_{[C_u \neq C_k]} \exp(-\gamma d_{uk}^2(W_{\mathcal{D}^{\setminus i}})))}{\sum_k \mathbb{I}_{[C_u = C_k, i \neq k]} \exp(-\gamma d_{ik}^2(W_{\mathcal{D}})) + \alpha \sum_k \mathbb{I}_{[C_u \neq C_k]} \exp(-\gamma d_{uk}^2(W_{\mathcal{D}}))} \Bigg|
\end{aligned}
\tag{42}
$$

Similar to the Eq. 28, we derive that

$$\|\Delta_{Z^{(l)}}\|_F \leq \rho \|U\|_2 \left\| \tilde{Z}^{(l-1)} W_{\mathcal{D}\backslash i}^{(l)} - Z^{(l-1)} W_{\mathcal{D}}^{(l)} \right\|_F$$

$$\leq \rho (1+\alpha)^{-1} \beta_A \left( \|\Delta_{Z^{(l-1)}}\|_F \|W_{\mathcal{D}\backslash i}^{(l)}\|_F + \|Z^{(l-1)}\|_F \|W_{\mathcal{D}\backslash i}^{(l)} - W_{\mathcal{D}}^{(l)}\|_F \right)$$

$$\leq \rho (1+\alpha)^{-1} \beta_A \left( \beta_{W_{\mathcal{D}}^{(l)}} \|\Delta_{Z^{(l-1)}}\|_F + \|W_{\mathcal{D}\backslash i}^{(l)} - W_{\mathcal{D}}^{(l)}\|_F \|Z^{(l-1)}\|_F \right)$$

$$\leq \rho^l \left(\frac{\beta_A}{1+\alpha}\right)^l \prod_{t=1}^l \beta_{W_{\mathcal{D}\backslash i}^{(t)}} \|\Delta_Z\|_F + \sum_{j=0}^l \left( \rho^l \left(\frac{\beta_A}{1+\alpha}\right)^l \prod_{t=1+j}^l \beta_{W_{\mathcal{D}\backslash i}^{(t)}} \prod_{t=1}^j \|W_{\mathcal{D}\backslash i}^{(j)} - W_{\mathcal{D}}^{(j)}\|_F \|Z\|_F \right)$$

$$\leq \eta \left(\frac{\rho \beta_A}{1+\alpha}\right)^l \sum_{j=0}^l \left( \prod_{t=1+j}^l \beta_{W_{\mathcal{D}\backslash i}^{(t)}} \prod_{t=1}^j \|W_{\mathcal{D}\backslash i}^{(j)} - W_{\mathcal{D}}^{(j)}\|_F \right).$$

(43)

And eventually, we obtain the worst-case difference as

$$E_1 \triangleq \sup_{\{u,v\}} \left| d_{uv}^2(W_{\mathcal{D}\backslash i}) - d_{uv}^2(W_{\mathcal{D}}) \right|$$

$$\leq \sup_{\{u,v\}} \left| \left(\frac{4}{h^2} + \frac{2\epsilon^4}{h^4}\right) \left( 2\Delta_{G_1}^4 + 2\Delta_{G_2}^4 + n(\Delta_{G_1} + \Delta_{G_2})^2 + \frac{\epsilon}{\sqrt{n}}(\Delta_{G_1} + \Delta_{G_2}) \right) \right|$$ (44)

$$\leq \left(\frac{4}{h^2} + \frac{2\epsilon^4}{h^4}\right) \left( 2\Delta_{G_1}^4 + 2\Delta_{G_2}^4 + n(\Delta_{G_1} + \Delta_{G_2})^2 + \frac{\epsilon}{\sqrt{n}}(\Delta_{G_1} + \Delta_{G_2}) \right)$$

where $\Delta_{G_i} = \eta \left(\frac{\rho \beta_A}{1+\alpha}\right)^l \sum_{j=0}^l \left( \prod_{t=1+j}^l \beta_{W_{\mathcal{D}\backslash i}^{(t)}} \prod_{t=1}^j \|W_{\mathcal{D}\backslash i}^{(j)} - W_{\mathcal{D}}^{(j)}\|_F \right)$ for all $i = 1, 2$. The worst-case scenario for the logarithmic term would be when the numerator is maximized, and the denominator is minimized, or vice versa. Let's denote this worst-case ratio as $E_2$:

$$E_2 \triangleq \sup_u \left| \log \frac{\sum_k \mathbb{I}_{[C_u=C_k, u\neq k]} \exp(-\gamma d_{uk}^2(W_{\mathcal{D}\backslash i})) + \alpha \sum_k \mathbb{I}_{[C_u \neq C_k]} \exp(-\gamma d_{uk}^2(W_{\mathcal{D}\backslash i}))}{\sum_k \mathbb{I}_{[C_u=C_k, i\neq k]} \exp(-\gamma d_{ik}^2(W_{\mathcal{D}})) + \alpha \sum_k \mathbb{I}_{[C_u \neq C_k]} \exp(-\gamma d_{uk}^2(W_{\mathcal{D}}))} \right|$$ (45)

Based on Lemma 4, the exponential term in Eq. 45 has the following inequality:

$$\exp(-\gamma d_{uk}^2(W_{\mathcal{D}\backslash i})) \leq \exp\left( -\gamma(d_{uk}^2(W_{\mathcal{D}\backslash i}) - d_{uk}^2(W_{\mathcal{D}})) \right) / \exp(-\gamma d_{uk}^2(W_{\mathcal{D}}))$$

$$\leq \left( 1 - \gamma\left( d_{uk}^2(W_{\mathcal{D}\backslash i}) - d_{uk}^2(W_{\mathcal{D}}) \right) + \frac{\gamma^2}{4}\left( d_{uk}^2(W_{\mathcal{D}\backslash i}) - d_{uk}^2(W_{\mathcal{D}}) \right)^2 \right) \cdot e^{\gamma d_{uk}^2(W_{\mathcal{D}})}$$

$$\leq \left( 1 + \gamma\left| d_{uk}^2(W_{\mathcal{D}\backslash i}) - d_{uk}^2(W_{\mathcal{D}}) \right| + \frac{\gamma^2}{4}\left| d_{uk}^2(W_{\mathcal{D}\backslash i}) - d_{uk}^2(W_{\mathcal{D}}) \right|^2 \right) \cdot e^{\gamma d_{uk}^2(W_{\mathcal{D}})}$$

$$= \left( 1 + \frac{\gamma}{2}\left| d_{uk}^2(W_{\mathcal{D}\backslash i}) - d_{uk}^2(W_{\mathcal{D}}) \right| \right)^2 \cdot e^{\gamma d_{uk}^2(W_{\mathcal{D}})}$$

$$\leq \left( 1 + \frac{\gamma}{2}\left(\frac{4}{h^2} + \frac{2\epsilon^4}{h^4}\right) \left( 2\Delta_{G_1}^4 + 2\Delta_{G_2}^4 + n(\Delta_{G_1} + \Delta_{G_2})^2 + \frac{\epsilon}{\sqrt{n}}(\Delta_{G_1} + \Delta_{G_2}) \right) \right)^2 \cdot e^{\gamma d_{uk}^2(W_{\mathcal{D}})}$$

(46)

For arbitrary two graphs $u, v$, assume their pairwise squared distance $d_{u,v}^2(W_{\mathcal{D}}) \leq \tau$. Denote $\theta_u^+$ as the number of other graphs with the same label as graph $u$, and $\theta_u^-$ as the number of other graphs with a different label from graph $u$, we then rewrite the expression for $E_2$:

$$
\begin{aligned}
E_2 \leq \sup_u \Bigg| \Bigg[ & \gamma\tau + \log(\theta_u^+ + \alpha\theta_u^-) \\
& + 2\log\left(1 + \frac{\gamma}{2}\left(\frac{4}{h^2} + \frac{2\epsilon^4}{h^4}\right)\left(2\Delta_{G_1}^4 + 2\Delta_{G_2}^4 + n(\Delta_{G_1} + \Delta_{G_2})^2 + \frac{\epsilon}{\sqrt{n}}(\Delta_{G_1} + \Delta_{G_2})\right)\right) \Bigg] \\
& \hspace{6cm} + \gamma\tau - \log(\theta_u^+ + \alpha\theta_u^-) \Bigg| \\
= 2\gamma\tau + 2\log&\left(1 + \frac{\gamma}{2}\left(\frac{4}{h^2} + \frac{2\epsilon^4}{h^4}\right)\left(2\Delta_{G_1}^4 + 2\Delta_{G_2}^4 + n(\Delta_{G_1} + \Delta_{G_2})^2 + \frac{\epsilon}{\sqrt{n}}(\Delta_{G_1} + \Delta_{G_2})\right)\right)
\end{aligned}
\tag{47}
$$

By combining Eq. 44, the supremum of the entire expression would be

$$
\sup_{\{u,v\}} |l(W_{\mathcal{D}}, Z_u, Z_v) - l(W_{\mathcal{D}\backslash i}, Z_u, Z_v)| \leq \gamma E_1 + E_2 \tag{48}
$$

$$
\leq \gamma\left(\frac{4}{h^2} + \frac{2\epsilon^4}{h^4}\right)\left(2\Delta_{G_1}^4 + 2\Delta_{G_2}^4 + n(\Delta_{G_1} + \Delta_{G_2})^2 + \frac{\epsilon}{\sqrt{n}}(\Delta_{G_1} + \Delta_{G_2})\right) +
$$

$$
+ 2\gamma\tau + 2\log\left(1 + \frac{\gamma}{2}\left(\frac{4}{h^2} + \frac{2\epsilon^4}{h^4}\right)\left(2\Delta_{G_1}^4 + 2\Delta_{G_2}^4 + n(\Delta_{G_1} + \Delta_{G_2})^2 + \frac{\epsilon}{\sqrt{n}}(\Delta_{G_1} + \Delta_{G_2})\right)\right)
$$

$$
= 4\gamma\left(\frac{2}{h^2} + \frac{\epsilon^4}{h^4}\right)\left(2\Delta_G^4 + 2n\Delta_G^2 + \frac{\epsilon}{\sqrt{n}}\Delta_G\right) + 2\gamma\tau + 2\log\left(1 + \gamma\left(\frac{2}{h^2} + \frac{\epsilon^4}{h^4}\right)\left(2\Delta_G^4 + 2n\Delta_G^2 + \frac{\epsilon}{\sqrt{n}}\Delta_G\right)\right)
$$

$$
\triangleq \omega
$$

where $\Delta_G = \eta\left(\frac{\rho\beta_A}{1+\alpha}\right)^l \sum_{j=0}^l \left(\prod_{t=1+j}^l \beta_{W_{\mathcal{D}\backslash i}^{(t)}} \prod_{t=1}^j \|W_{\mathcal{D}\backslash i}^{(j)} - W_{\mathcal{D}}^{(j)}\|_F\right)$. $\qquad\square$

# G  THEORETICAL AND EMPIRICAL COMPLEXITY

## G.1  COMPLEXITY ANALYSIS

**Proof for Theorem 3**

*Proof.* For the embedding propagation step, it requires $\mathcal{O}(Lmd)$ runtimes. In the kernel mapping step, we get a pairwise kernel value by calculating the generalized MMD. With a single kernel, it needs $\mathcal{O}(n^2d)$ runtimes. Consider a kernel family $\mathcal{K}$ with a size of $\kappa$, the complexity of this part is $\mathcal{O}(\kappa n^2 d)$. Overall, the one pairwise MMD-GK is computed in $\mathcal{O}(Lmd + \kappa n^2 d)$ and for $N$ graphs, the complexity of all pairs is $\mathcal{O}(NLmd + N^2\kappa n^2 d)$. $\qquad\square$

**Proof for Theorem 4**

*Proof.* For the embedding propagation step, it requires $\mathcal{O}(Lnd^2 + Lmd)$ runtimes. Other things being equal to MMD-GK, one pairwise Deep MMD-GK is therefore computed in $\mathcal{O}(Lnd^2 + Lmd + \kappa n^2 d)$. For $N$ graphs, the complexity of all pairs is $\mathcal{O}(NLnd^2 + NLmd + N^2\kappa n^2 d)$. $\qquad\square$

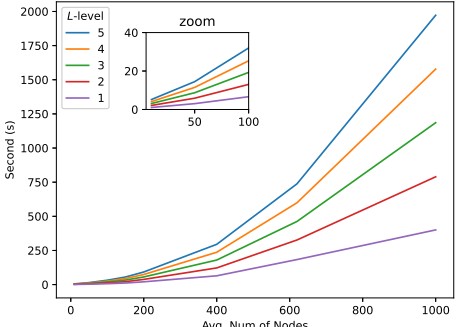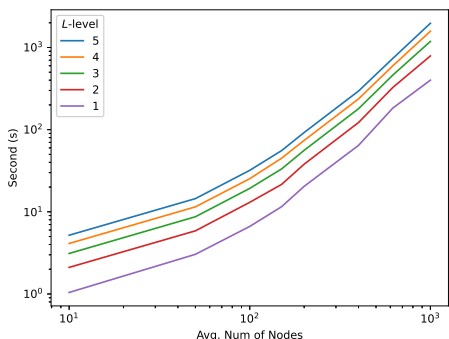

Figure 3: Runtimes Performance of MMD-GK computation with 1000 synthetic graphs and one RBF kernel $|\mathcal{K}|= 1$. We report both normal and log-scale runtime with different levels ($L$) of node features, where $L \in \{1, 2, 3, 4, 5\}$.

## G.2    RUNTIME ANALYSIS

Following the setting of Togninalli et al. (2019), we conducted a simulation using a consistent number of graphs while varying the average node number per graph, to assess the performance of our MMD-GK approach with respect to the average number of nodes. Specifically, we generated 1000 synthetic graphs with the same node feature's dimension $d = 20$. For each graph, we generate the number of nodes ($n$) based on a normal distribution centered on an average node number, and the number of edges $m$ is controlled between $n$ and $5n$. For each level of node features $L \in \{1, 2, 3, 4, 5\}$, we applied the MMD-GK with a single Radial Basis Function (RBF) kernel (i.e. $|\mathcal{K}|= 1$) to measure and compare runtimes over the different average number of nodes $n$. From Figure 3, the runtimes increase the runtimes increase with the square of the number of nodes $n^2$, which is modulated by the number of levels $L$. In practical terms, the graph on normal scales suggests a linear trend, especially for graphs with a smaller number of nodes. As the number of nodes increases, the complexity term involving $n^2$ might become more dominant, which could explain the superlinear growth observed, particularly at higher levels of $L$. This is consistent with the theoretical complexity, which posits that as $L$ and $n$ increase, the runtimes should increase correspondingly. The log scale also helps to visualize the dramatic impact of the $n^2$ term as the number of nodes grows.

We also compare our approaches to four graph kernels on the synthetic graphs. For the Graphlet kernel, $k = 5$; for the Weisfeiler-Lehman Subtree kernel, $h = 5$; for our MMD-GK and Deep MMD-GK, $L = h$, $\kappa = 1$ and $d = 1$. $m$ is held constant across the kernels. All methods are performed ten times. Table 4 indicates that our approaches are competitive with the existing graph kernel methods.

## H    EXPERIMENT SETTINGS

**Configuration**    DHFR, BZR, and PROTEINS have node attributes whereas MUTAG and PTC_FM only contain node labels. For GNN-related methods, we transform the categorical node label to one-hot embedding. We evaluate our approaches exhaustively on all possible graph distributions for each dataset in Table 5, i.e. for datasets without node labels, $Z = Y$; and for datasets with node labels, $Z = X$ or $Z = Y$ or $Z = (X, Y)$. To avoid over-smoothing, we set the number of level $L \in \{1, 2, 3, \cdots, 10\}$ for MMD-GKs and $L \in \{1, 2, 3, 4\}$ for Deep MMD-GKs and adopt a Gaussian kernel family with bandwidth $h \in \{$1e-2, 1e-1, 1e0, 1e1, 1e2$\}$. For distance-to-similarity, we explore scenarios when $\gamma \in \{$1e-1, 1e0, 1e1$\}$. For

Table 4: Total computation time for 10 runs on 1000 synthetic graphs with an average of 20 nodes and 50 edges.

| Graph Kernel | Total Runtime |
|---|---|
| Graphlet | 120.4s |
| Shortest Path | 30.3.s |
| Weisfeiler-Lehman Subtree | 26.4 s |
| Wasserstein Weisfeiler-Lehman | 211.0s |
| MMD-GK | 29.2s |
| Deep MMD-GK | 31.2s |

Table 5: Description of the benchmark datasets DHFR, BZR, MUTAG, PTC_FM and PROTEINS

| Dataset | DHFR | BZR | MUTAG | PTC_FM | PROTEINS |
|---|---|---|---|---|---|
| Num. of graphs | 467 | 405 | 188 | 349 | 1113 |
| Num. of graph labels | 2 | 2 | 2 | 2 | 2 |
| Dim. of node attributes | 3 | 3 | / | / | 29 |
| Num. of node labels | 9 | 10 | 7 | 18 | 3 |
| Avg. number of nodes | 42.43 | 35.75 | 17.93 | 14.11 | 39.06 |
| Avg. number of edges | 44.54 | 38.36 | 19.79 | 14.48 | 72.82 |
| Label Proportion | 461/295 | 319/86 | 125/63 | 206/143 | 663/450 |

Deep MMD-GK, we implement the algorithm in Pytorch with $\sigma$ set as LeakyRelu activation and $\lambda$ fixed to be $0.33$ which means there are around one-third of pairs selected as negative samples in unsupervised contrastive loss. All Deep MMD-GKs are trained for 300 epochs with a batch size of 128, using Adam optimizer (Kingma & Ba, 2015) at 1e-2 initial learning rate. For graph clustering, we adopt MMD-GK and unsupervised Deep MMD-GK as the precomputed affinity matrix to run Spectral Clustering (Pedregosa et al., 2011). For graph classification, we use a binary C-SVM (Chang & Lin, 2011) and report the average of test accuracy across the 10 folds within the cross-validation. The parameter C for each fold is tuned from $\{1e0, 1e1, 1e2\}$. For the graph kernel baselines, we report the best performances across all parameters: depth of the subtree $h \in \{1, 2, 3, 4, 5\}$ for WL Kernel; graphlet size $k \in \{3, 4, 5, 6\}$ for Graphlet Kernel. For other baselines, we set their parameters according to their original papers.

**Evaluation Metrics** For the graph classification task, we use the accuracy which is defined as a ratio of the correct predictions. For the graph clustering task, we use the following scores to evaluate the result: 1) ACC (Accuracy): Represents the proportion of the total number of predictions that are correct; 2) NMI (Normalized Mutual Information): Measures the mutual information of the true and predicted clusterings, normalized to have values between 0 (no mutual information, or independent labelings) and 1 (perfect correlation); ARI (Adjusted Rand Index): Evaluates the similarity of two clusterings, adjusted for chance. ARI ranges between -1 and 1, with 1 indicating a perfect match, 0 indicating random labeling, and negative values indicating a bad match.

# I  PARAMETER SENSITIVITY

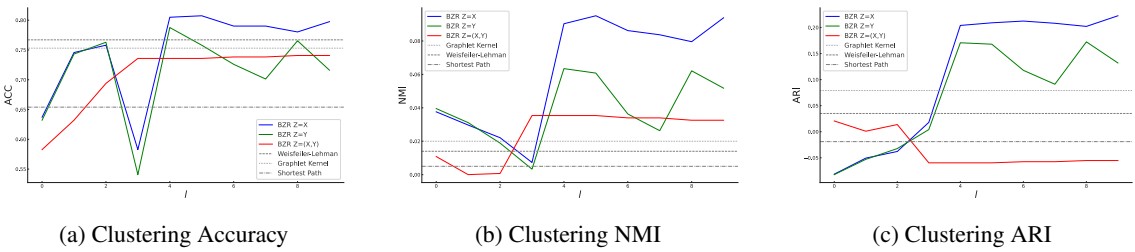

(a) Clustering Accuracy          (b) Clustering NMI          (c) Clustering ARI

Figure 4: Parameter Sensitivity of MMD-GK on the dataset BZR; comparison of scores with change in level $l$ (x-axis) and graph distribution $\mathbb{P}_Z$ ($Z = X$ in blue, $Z = Y$ in green and $Z = (X,Y)$ in red).

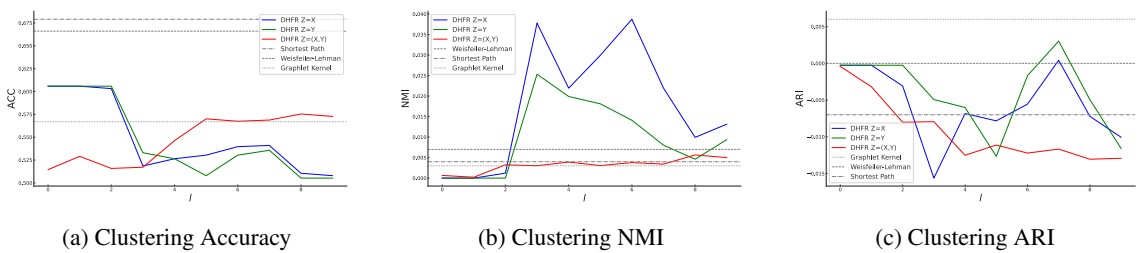

(a) Clustering Accuracy          (b) Clustering NMI          (c) Clustering ARI

Figure 5: Parameter Sensitivity of MMD-GK on the dataset DHFR; Comparison of scores with change in level $l$ (x-axis) and graph distribution $\mathbb{P}_Z$ ($Z = X$ in blue, $Z = Y$ in green and $Z = (X,Y)$ in red).

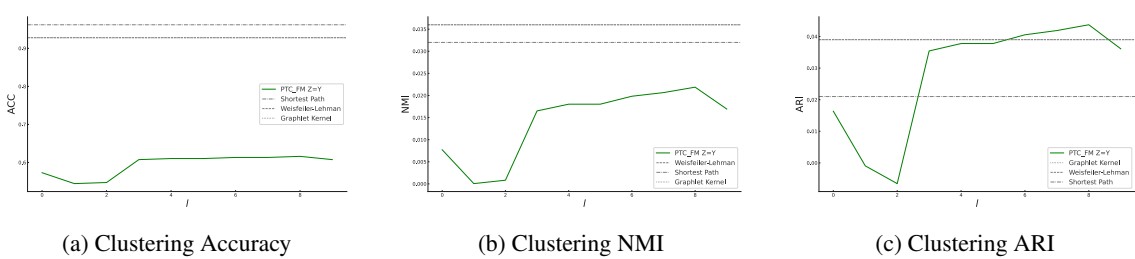

(a) Clustering Accuracy          (b) Clustering NMI          (c) Clustering ARI

Figure 6: Parameter Sensitivity of MMD-GK on the dataset PTC_FM; Comparison of scores with change in level $l$ (in x-axis).

## J  DEEP MMD-GK CONVERGENCE

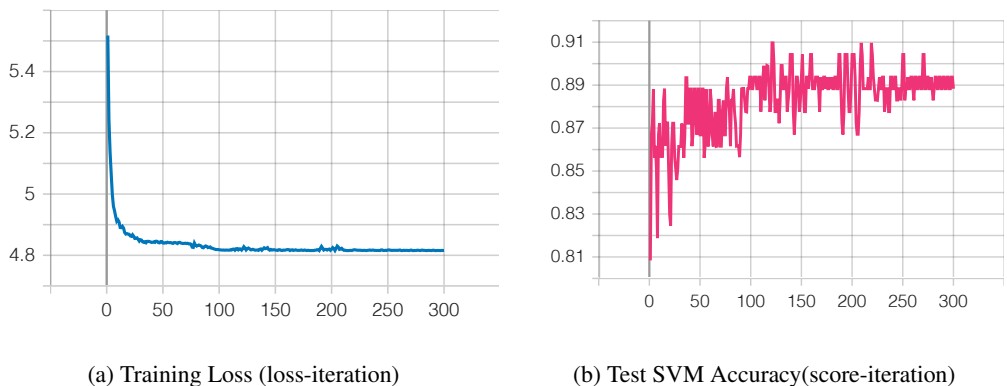

(a) Training Loss (loss-iteration)  (b) Test SVM Accuracy(score-iteration)

Figure 7: Convergence in training supervised Deep MMD-GK on the benchmark dataset MUTAG.

## K  ADDITIONAL RESULTS

In this section, we present detailed tables showcasing the performance evaluation of both supervised and unsupervised models on two datasets: MUTAG and PTC_FM. These tables provide a comprehensive breakdown of the results, factoring in various parameters such as loss type (SCL, KL, UCL), levels ($l$-level), and tuning parameters ($\alpha$ and $\gamma$). Each table is accompanied by a caption that highlights the best performance metrics in bold, aiding in the quick identification of the most effective parameter configurations.

**Performance Trends**  Increasing $\gamma$ values generally lead to better performance in supervised models on the MUTAG dataset, suggesting that this parameter positively influences the model's ability to learn from this particular dataset. The $l$-level seems to have a less consistent impact, but higher levels do not always equate to better performance, indicating a complex interaction with the dataset's characteristics. Supervised models outperform unsupervised models in most configurations, which could be due to the additional guidance provided by labeled data.

**Model Behavior**  For graph classification, simpler models (lower $l$-levels) may perform better, whereas, for graph clustering, more complex ones with KL loss (higher $l$-levels) may perform better. The varying performance across different $l$-levels and hyperparameter settings for the two datasets (MUTAG and PTC_FM) implies that these models respond differently to distinct data characteristics, such as graph distribution and class imbalances.

Table 6: Supervised Model Evaluation on MUTAG. The best performance is highlighted in **bold**.

| Loss | $l$-level | $\alpha$ | $\gamma$ | SVC |
|------|-----------|----------|----------|-----|
| SCL | 1 | 0.1 | 0.1 | $0.665 \pm 0.009$ |
| SCL | 1 | 0.1 | 1 | $0.808 \pm 0.060$ |
| SCL | 1 | 0.1 | 10 | $0.883 \pm 0.054$ |
| SCL | 1 | 1 | 0.1 | $0.665 \pm 0.009$ |
| SCL | 1 | 1 | 1 | $0.830 \pm 0.041$ |
| SCL | 1 | 1 | 10 | $0.904 \pm 0.032$ |
| SCL | 1 | 10 | 0.1 | $0.670 \pm 0.011$ |
| SCL | 1 | 10 | 1 | $0.835 \pm 0.049$ |
| SCL | 1 | 10 | 10 | $\mathbf{0.915 \pm 0.065}$ |
| SCL | 2 | 0.1 | 0.1 | $0.681 \pm 0.028$ |
| SCL | 2 | 0.1 | 10 | $0.909 \pm 0.054$ |
| SCL | 2 | 0.1 | 1 | $0.819 \pm 0.025$ |
| SCL | 2 | 1 | 0.1 | $0.665 \pm 0.009$ |
| SCL | 2 | 1 | 1 | $0.814 \pm 0.040$ |
| SCL | 2 | 1 | 10 | $0.893 \pm 0.029$ |
| SCL | 2 | 10 | 0.1 | $0.670 \pm 0.009$ |
| SCL | 2 | 10 | 1 | $0.819 \pm 0.025$ |
| SCL | 2 | 10 | 10 | $0.888 \pm 0.040$ |
| SCL | 3 | 0.1 | 0.1 | $0.676 \pm 0.009$ |
| SCL | 3 | 0.1 | 1 | $0.825 \pm 0.038$ |
| SCL | 3 | 0.1 | 10 | $0.857 \pm 0.035$ |
| SCL | 3 | 1 | 0.1 | $0.686 \pm 0.0169$ |
| SCL | 3 | 1 | 1 | $0.819 \pm 0.054$ |
| SCL | 3 | 1 | 10 | $0.840 \pm 0.031$ |
| SCL | 3 | 10 | 0.1 | $0.713 \pm 0.039$ |
| SCL | 3 | 10 | 1 | $0.851 \pm 0.043$ |
| SCL | 3 | 10 | 10 | $0.888 \pm 0.085$ |
| SCL | 4 | 0.1 | 0.1 | $0.734 \pm 0.043$ |
| SCL | 4 | 0.1 | 1 | $0.841 \pm 0.075$ |
| SCL | 4 | 0.1 | 10 | $0.745 \pm 0.044$ |
| SCL | 4 | 1 | 0.1 | $0.719 \pm 0.027$ |
| SCL | 4 | 1 | 1 | $0.792 \pm 0.030$ |
| SCL | 4 | 1 | 10 | $0.745 \pm 0.024$ |
| SCL | 4 | 10 | 0.1 | $0.857 \pm 0.063$ |
| SCL | 4 | 10 | 1 | $0.872 \pm 0.061$ |
| SCL | 4 | 10 | 10 | $0.761 \pm 0.022$ |

Table 7: Unsupervised Model Evaluation on MUTAG. The best performance across each loss is highlighted in **bold**.

| Loss | $l$-level | $\alpha$ | $\gamma$ | ACC | NMI | ARI | SVC |
|------|-----------|----------|----------|-----|-----|-----|-----|
| KL | 1 | – | 0.1 | 0.713 | 0.147 | 0.177 | $0.665 \pm 0.009$ |
| KL | 1 | – | 1 | 0.718 | 0.152 | 0.186 | $0.873 \pm 0.026$ |
| KL | 1 | – | 10 | 0.750 | 0.172 | 0.244 | $0.899 \pm 0.019$ |
| KL | 2 | – | 0.1 | 0.702 | 0.148 | 0.159 | $0.873 \pm 0.026$ |
| KL | 2 | – | 1 | 0.723 | 0.157 | 0.195 | $0.893 \pm 0.029$ |
| KL | 2 | – | 10 | 0.723 | 0.152 | 0.186 | $0.873 \pm 0.026$ |
| KL | 3 | – | 0.1 | 0.745 | 0.249 | 0.236 | $0.665 \pm 0.009$ |
| KL | 3 | – | 1 | 0.840 | 0.459 | 0.461 | $0.893 \pm 0.009$ |
| KL | 3 | – | 10 | 0.840 | 0.432 | 0.454 | $0.894 \pm 0.021$ |
| KL | 4 | – | 0.1 | **0.846** | **0.468** | **0.476** | $0.713 \pm 0.009$ |
| KL | 4 | – | 1 | 0.819 | 0.440 | 0.405 | $0.883 \pm 0.009$ |
| KL | 4 | – | 10 | 0.840 | 0.459 | 0.461 | $\mathbf{0.910 \pm 0.047}$ |
| UCL | 1 | 0.1 | 0.1 | 0.713 | 0.141 | 0.177 | $0.665 \pm 0.009$ |
| UCL | 1 | 0.1 | 1 | 0.713 | 0.141 | 0.177 | $0.819 \pm 0.054$ |
| UCL | 1 | 0.1 | 10 | 0.750 | 0.172 | 0.244 | $0.893 \pm 0.058$ |
| UCL | 1 | 1 | 0.1 | 0.713 | 0.141 | 0.177 | $0.665 \pm 0.009$ |
| UCL | 1 | 1 | 1 | 0.713 | 0.141 | 0.177 | $0.83 \pm 0.069$ |
| UCL | 1 | 1 | 10 | 0.745 | 0.166 | 0.233 | $\mathbf{0.910 \pm 0.026}$ |
| UCL | 1 | 10 | 0.1 | 0.713 | 0.147 | 0.177 | $0.665 \pm 0.009$ |
| UCL | 1 | 10 | 1 | 0.713 | 0.141 | 0.177 | $0.824 \pm 0.049$ |
| UCL | 1 | 10 | 10 | 0.745 | 0.166 | 0.233 | $\mathbf{0.910 \pm 0.045}$ |
| UCL | 2 | 0.1 | 0.1 | 0.713 | 0.167 | 0.177 | $0.665 \pm 0.009$ |
| UCL | 2 | 0.1 | 1 | 0.729 | 0.172 | 0.205 | $0.814 \pm 0.079$ |
| UCL | 2 | 0.1 | 10 | 0.750 | 0.181 | 0.240 | $0.878 \pm 0.068$ |
| UCL | 2 | 1 | 0.1 | 0.734 | 0.167 | 0.214 | $0.665 \pm 0.009$ |
| UCL | 2 | 1 | 1 | 0.745 | 0.175 | 0.234 | $0.819 \pm 0.072$ |
| UCL | 2 | 1 | 10 | 0.750 | 0.164 | 0.241 | $0.888 \pm 0.048$ |
| UCL | 2 | 10 | 0.1 | 0.729 | 0.152 | 0.204 | $0.665 \pm 0.009$ |
| UCL | 2 | 10 | 1 | 0.713 | 0.167 | 0.176 | $0.815 \pm 0.117$ |
| UCL | 2 | 10 | 10 | 0.750 | 0.170 | 0.243 | $\mathbf{0.910 \pm 0.052}$ |
| UCL | 3 | 0.1 | 0.1 | 0.766 | 0.207 | 0.276 | $0.665 \pm 0.009$ |
| UCL | 3 | 0.1 | 1 | 0.750 | 0.207 | 0.246 | $0.830 \pm 0.046$ |
| UCL | 3 | 0.1 | 10 | 0.771 | 0.174 | 0.274 | $0.856 \pm 0.039$ |
| UCL | 3 | 1 | 0.1 | 0.761 | 0.221 | 0.268 | $0.665 \pm 0.009$ |
| UCL | 3 | 1 | 1 | 0.750 | 0.233 | 0.246 | $0.846 \pm 0.046$ |
| UCL | 3 | 1 | 10 | 0.739 | 0.211 | 0.217 | $0.861 \pm 0.054$ |
| UCL | 3 | 10 | 0.1 | 0.739 | 0.175 | 0.225 | $0.665 \pm 0.009$ |
| UCL | 3 | 10 | 1 | 0.761 | 0.234 | 0.268 | $0.798 \pm 0.098$ |
| UCL | 3 | 10 | 10 | 0.761 | 0.237 | 0.255 | $0.836 \pm 0.049$ |
| UCL | 4 | 0.1 | 0.1 | 0.787 | 0.320 | 0.324 | $0.676 \pm 0.018$ |
| UCL | 4 | 0.1 | 1 | **0.798** | 0.252 | **0.345** | $0.824 \pm 0.027$ |
| UCL | 4 | 0.1 | 10 | 0.787 | 0.252 | 0.314 | $0.761 \pm 0.035$ |
| UCL | 4 | 1 | 0.1 | 0.793 | 0.249 | 0.328 | $0.670 \pm 0.009$ |
| UCL | 4 | 1 | 1 | 0.787 | 0.248 | 0.312 | $0.803 \pm 0.029$ |
| UCL | 4 | 1 | 10 | 0.787 | 0.247 | 0.323 | $0.740 \pm 0.019$ |
| UCL | 4 | 10 | 0.1 | 0.771 | 0.266 | 0.291 | $0.692 \pm 0.009$ |
| UCL | 4 | 10 | 1 | 0.787 | **0.338** | 0.327 | $0.819 \pm 0.059$ |
| UCL | 4 | 10 | 10 | **0.798** | 0.244 | **0.345** | $0.868 \pm 0.034$ |

Table 8: Supervised Model Evaluation on PTC_FM. The best performance is highlighted in **bold**.

| Loss | $l$-level | $\alpha$ | $\gamma$ | SVC |
|------|-----------|----------|----------|-----|
| SCL | 1 | 1 | 0.1 | $0.639 \pm 0.014$ |
| SCL | 1 | 1 | 1 | $0.659 \pm 0.029$ |
| SCL | 1 | 1 | 10 | $0.613 \pm 0.027$ |
| SCL | 1 | 10 | 10 | $0.633 \pm 0.031$ |
| SCL | 1 | 10 | 1 | $\mathbf{0.667 \pm 0.032}$ |
| SCL | 2 | 1 | 0.1 | $0.648 \pm 0.025$ |
| SCL | 2 | 1 | 1 | $0.662 \pm 0.043$ |
| SCL | 2 | 1 | 10 | $0.633 \pm 0.030$ |
| SCL | 2 | 10 | 1 | $0.665 \pm 0.045$ |
| SCL | 2 | 10 | 10 | $0.605 \pm 0.019$ |
| SCL | 3 | 1 | 1 | $0.659 \pm 0.040$ |
| SCL | 3 | 1 | 10 | $0.613 \pm 0.027$ |
| SCL | 3 | 10 | 1 | $0.665 \pm 0.026$ |
| SCL | 3 | 10 | 10 | $0.619 \pm 0.047$ |
| SCL | 4 | 1 | 1 | $0.662 \pm 0.047$ |
| SCL | 4 | 1 | 10 | $0.625 \pm 0.032$ |
| SCL | 4 | 10 | 1 | $0.656 \pm 0.026$ |
| SCL | 4 | 10 | 10 | $0.610 \pm 0.029$ |

Table 9: Unsupervised Model Evaluation on PTC_FM. The best performance across each loss is highlighted in **bold**.

| Loss | $l$-level | $\alpha$ | $\gamma$ | ACC | NMI | ARI | SVC |
|------|-----------|----------|----------|-----|-----|-----|-----|
| KL | 1 | 1 | 1 | 0.567 | 0.009 | 0.014 | $\mathbf{0.659 \pm 0.032}$ |
| KL | 1 | 1 | 10 | 0.625 | 0.076 | 0.035 | $0.610 \pm 0.023$ |
| KL | 2 | 1 | 1 | 0.619 | 0.022 | 0.042 | $0.653 \pm 0.028$ |
| KL | 2 | 1 | 10 | 0.616 | 0.044 | 0.025 | $0.636 \pm 0.032$ |
| KL | 3 | 1 | 1 | 0.599 | 0.014 | 0.030 | $0.650 \pm 0.006$ |
| KL | 3 | 1 | 10 | 0.648 | **0.073** | 0.065 | $0.653 \pm 0.048$ |
| KL | 4 | 1 | 1 | **0.642** | 0.042 | **0.066** | $0.633 \pm 0.019$ |
| KL | 4 | 1 | 10 | 0.599 | 0.021 | 0.007 | $0.619 \pm 0.027$ |
| UCL | 1 | 1 | 1 | **0.630** | 0.031 | **0.057** | $\mathbf{0.668 \pm 0.028}$ |
| UCL | 1 | 1 | 10 | 0.625 | **0.052** | 0.035 | $0.616 \pm 0.043$ |
| UCL | 2 | 1 | 1 | 0.625 | 0.029 | 0.054 | $0.665 \pm 0.053$ |
| UCL | 2 | 1 | 10 | 0.607 | 0.040 | 0.016 | $0.636 \pm 0.014$ |
| UCL | 3 | 1 | 1 | 0.616 | 0.024 | 0.046 | $0.650 \pm 0.028$ |
| UCL | 3 | 1 | 10 | 0.599 | 0.021 | 0.007 | $0.607 \pm 0.021$ |
| UCL | 4 | 1 | 1 | 0.625 | 0.028 | 0.053 | $0.651 \pm 0.017$ |
| UCL | 4 | 1 | 10 | 0.599 | 0.021 | 0.007 | $0.630 \pm 0.013$ |