# OpenReview forum: "MMD Graph Kernel: Effective Metric Learning for Graphs via Maximum Mean Discrepancy"
_ICLR.cc/2024/Conference — ICLR 2024 spotlight_

### Official Review · Reviewer_CP9G · 2023-10-29

**Soundness:** 4 excellent
**Presentation:** 3 good
**Contribution:** 4 excellent
**Rating:** 8
**Confidence:** 5

**Summary:**

The paper introduced a class of MMD based graph kernels. It also introduced a class of deep MMD graph kernels that have learnable parameters. Some theoretical analysis such as robustness have been done. The experiments of graph clustering and classification showed the effectiveness of the proposed methods.

**Strengths:**

1. Measuring the similarity between graphs is a practical and challenging problem.
2. The proposed shallow and deep MMD graph kernels are novel and interesting. The deep MMD graph kernel is learnable in both supervised and unsupervised ways.
3. The paper studied the robustness and generalization of the graph kernels.
4. In the experiments, the proposed deep MMD graph kernel is more effective than classical graph kernels and graph neural networks. The superior performance in clustering is quite impressive.
In sum, this is a good work with solid theoretical analysis and numerical evaluation. The unsupervised deep MMD graph kernel makes a breakthrough in the area.

**Weaknesses:**

The paper has the following weaknesses..
1. The motivation of the two unsupervised loss functions (eq.10 and eq.11) in Section 4 haven’t been sufficiently explained.
2. For graph clustering, GNN-based clustering method should be compared in Table 1.

**Questions:**

It would be great if the authors could improve the aforementioned weaknesses and answer the following questions.
1. In Definition 5, for the generalized MMD, how to efficiently determine $k$ in practice? It may have infinite choices.
2. In Theorem 1, it seems that if $h$ is close to zero, the bound is loose. How to avoid small $h$ in practice?
3. In unsupervised learning with loss functions (11) and (12), will model collapse (all $s_{ij}$ are zero or one) happen? This may be caused by zero or infinity weights in the deep model.
4. More explanation about the usefulness of (11) and (12) are required.
5. In Section 4.2, suppose $\beta$ can be computed, how to obtain the stability parameter $\omega$?
6. In Table 1, the NMIs and ARIs given by all methods on datasets DHFR and PTC are very low. What is the possible reason?

---

> ### Author Response · Authors · 2023-11-22
> **Rebuttal for Reviewer CP9G**
>
> **W1:** The motivation of the two unsupervised loss functions (eq.10 and eq.11) in Section 4 haven’t been sufficiently explained.
>
> **Response:** Thanks for your suggestion. We would further explain the motive behind the unsupervised loss function.
> - Supervised Contrastive Loss (SCL) equation is designed to enhance the model's ability to learn from graph labels when they are available. By adopting the supervised version of the InfoNEC loss function SCL focuses on maximizing the agreement between embeddings of graphs that share the same label (indicated by $C_i = C_j$). The equation reflects this by encouraging similarity (high $s\_{ij}$) between graphs of the same class while penalizing similarity between graphs of different classes. The component $\alpha \sum\_{k}\mathbb{I}{[C\_i \ne C\_i]}s_{ik}$ in the denominator acts as a regularizing term to control the influence of dissimilar pairs.
> - Unsupervised Contrastive Loss (UCL) equation provides a mechanism for learning the MMD graph kernel when graph labels are not available. This loss function is a modification of SCL where the distinction between similar and dissimilar graph pairs is made based on their similarity rather than their labels. UCL focuses on selecting the most similar pairs ($S\_+$) and the least similar pairs ($S\_-$) within each training epoch. The loss function encourages the model to distinguish between these two sets in the absence of explicit labels.
>
> **W2:** For graph clustering, GNN-based clustering method should be compared in Table 1.
>
> **Response:** Thanks for your suggestion. Following your recommendation, we have updated the table to incorporate results from two notable GNN methods [1][2] for clustering. We use the two methods to obtain graph representations and then conduct spectral clustering. Our methods outperformed these two baselines.
>
> [1] You, Yuning, et al. "Graph contrastive learning with augmentations." NeurIPS 2020
> [2] Sun, Fan-Yun, et al. "InfoGraph: Unsupervised and Semi-supervised Graph-Level Representation Learning via Mutual Information Maximization." ICLR 2020.
>
> ---
>
> **Q1:** In Definition 5, for the generalized MMD, how to efficiently determine $k$ in practice? It may have infinite choices.
>
> **Response:** Thanks for your inquiry. A common way to determine the $k$ is to set the parameter range with limited sparse choices such as using an RBF kernel family with bandwidths like $[1e-1, 1e0, 1e1]$. This method provides a balance between computational efficiency and the ability to capture different scales of features in the data.
>
> **Q2:** In Theorem 1, it seems that if $h$ is close to zero, the bound is loose. How to avoid small $h$ in practice?
>
> **Response:** Thanks for bringing up this point. To address this issue in practice, the most straightforward way is avoid including a minimal (e.g. smaller than $1e-3$) bandwidth in the kernel family.
>
> **Q3:** In unsupervised learning with loss functions (11) and (12), will model collapse (all $s_{ij}$ are zero or one) happen? This may be caused by zero or infinity weights in the deep model.
>
> **Response:** Thanks for your question. In the case of UCL, employing a diverse set of negative samples (i.e., non-similar pairs) can prevent the model from trivially maximizing similarity scores. In practice, we found the non-similar pairs could ensure a representative sample of dissimilar pairs discourages the model from collapsing towards uniform similarity scores. For KL loss, we have implemented regularization techniques to prevent extreme values in the model weights.
>
> **Q4:** More explanation about the usefulness of (11) and (12) are required.
>
> **Response:** Thanks for your suggestions. Both UCL and KL serve crucial roles in our approach to unsupervised learning from unlabeled data. The primary benefit of UCL is that it facilitates the learning of discriminative features from a set of graphs. By minimizing the KL Divergence, the model learns to adjust its embeddings to better reflect the natural clustering of the data. We have added the explanation to Section 4 in the manuscript.
>
> **Q5:** In Section 4.2, suppose $\beta$ can be computed, how to obtain the stability parameter $w$ ?
>
> **Response:** Thanks for your query. We provide the explicit formulation for the stability parameter $\omega$ in Appendix F.

---

> > ### Author Response · Authors · 2023-11-22
> > **Further Clarification**
> >
> > **Q6:** In Table 1, the NMIs and ARIs given by all methods on datasets DHFR and PTC are very low. What is the possible reason?
> >
> > **Response:** Thanks for your inquiry. We think there are several reasons why NMI and ARI are very low, but ACC is relatively high (around 0.6-0.7). Firstly, both NMI and ARI are sensitive to the number and distribution of clusters. A low NMI or ARI score implies that the clustering groups discovered by the algorithm do not align well with the actual distribution of classes in the dataset. If the true clustering significantly varied cluster sizes (for DHFR 2, it's 1.6:1, and for PTC it's 1.9:1), NMI and ARI may give low scores even if the clusters identified by the algorithm are reasonably accurate. Also, in datasets with imbalanced class distributions, spectral clustering might correctly identify larger classes but fail in accurately clustering smaller ones. This would result in a decent ACC but poor NMI and ARI scores. For instance, the Shortest Path graph kernel produces a high ACC in PTC_FM, but the predicted clusters are highly imbalanced (size ratio is about 26:1).
> >
> > **Hope this response can solve your concerns. We thank the reviewer again for recognizing our work.**

---

> > ### Author Response · Authors · 2023-11-22
> > **Updated Table of Graph Clustering**
> >
> > We provide the updated graph clustering table for a clear illustration to your **W2**. Here, we bold our models that exceed the baseline and show the best score in italics for each dataset.
> >
> > |            Model           | Metric |     BZR     |     DHFR    |    MUTAG    |    PTC_FM   |   PROTEINS  |
> > |:--------------------------:|:------:|:-----------:|:-----------:|:-----------:|:-----------:|:-----------:|
> > |       Graphlet Kernel      |   ACC  |    0.753    |    0.567    |    0.771    |    0.928    |    0.622    |
> > |                            |   NMI  |    0.020    |    0.003    |    0.143    |    0.036    |    0.053    |
> > |                            |   ARI  |    0.079    |    0.006    |    0.234    |    0.039    |    0.027    |
> > |        Shortest Path       |   ACC  |    0.654    |    0.679    |    0.676    |   _0.962_   |    0.604    |
> > |                            |   NMI  |    0.005    |    0.004    |    0.211    |    0.032    |    0.027    |
> > |                            |   ARI  |    -0.019   |    -0.007   |    -0.318   |    0.021    |    0.041    |
> > |      Weisfeiler-Lehman     |   ACC  |    0.767    |    0.666    |    0.548    |    0.928    |    0.582    |
> > |                            |   NMI  |    0.014    |    0.007    |    0.136    |    0.036    |    0.021    |
> > |                            |   ARI  |    0.035    |    0.000    |    0.176    |    0.039    |    0.026    |
> > |          InfoGraph         |   ACC  |    0.735    |    0.658    |    0.726    |    0.620    |    0.732    |
> > |                            |   NMI  |    0.036    |    0.032    |    0.287    |    0.021    |    0.132    |
> > |                            |   ARI  |    0.050    |    0.005    |    0.199    |    0.046    |    0.124    |
> > |           GraphCL          |   ACC  |    0.729    |    0.652    |    0.732    |    0.621    |    0.728    |
> > |                            |   NMI  |    0.019    |    0.040    |    0.322    |    0.021    |    0.140    |
> > |                            |   ARI  |    0.035    |    0.003    |    0.234    |    0.034    |    0.115    |
> > |           MMD-GK           |   ACC  |  **0.807**  |    0.614    |    0.707    |    0.633    |    0.730    |
> > |                            |   NMI  |  **0.095**  |  **0.042**  |    0.141    |    0.034    |  **0.144**  |
> > |                            |   ARI  |  **0.232**  |    0.004    |    0.168    |  **0.060**  |  **0.207**  |
> > | Deep MMD-GK (unsupervised) |   ACC  | **_0.952_** | **_0.689_** | **_0.846_** |    0.647    | **_0.752_** |
> > |                            |   NMI  | **_0.757_** | **_0.182_** | **_0.468_** | **_0.076_** | **_0.333_** |
> > |                            |   ARI  | **_0.809_** | **_0.137_** | **_0.476_** | **_0.066_** | **_0.345_** |

---

> > ### Comment · Reviewer_CP9G · 2023-11-22
> >
> > Thanks a lot for the responses，especially the derivation of $\omega$ and results of new baselines. My concerns have been well addressed. I keep my score and raise the confidence level to 5.

---

> > > ### Author Response · Authors · 2023-11-22
> > > **Gratitude for Confidence Boost**
> > >
> > > We appreciate your acknowledgment of the recent updates and derivations presented. Your decision to raise the confidence level serves as a significant encouragement to our efforts. We remain committed to continuous improvement and are grateful for your constructive feedback, which has undoubtedly pushed our work forward.

---

### Official Review · Reviewer_avjd · 2023-10-30

**Soundness:** 3 good
**Presentation:** 3 good
**Contribution:** 3 good
**Rating:** 8
**Confidence:** 5

**Summary:**

This work proposed a class of (deep) MMD based graph kernels, coupled with theoretical results. The proposed kernels are applied to graph classification and graph clustering on a few benchmark datasets.

**Strengths:**

* The paper proposed a class of (deep) MMD based graph kernels (MMD-GKs). The novelty is high. Different from classical graph kernels that have fixed features, the proposed Deep MMD-GKs are learnable, with or without labels.
* The authors provided some theoretical analysis, which are useful and explainable.
* The deep MMD-GKs outperformed the baselines in supervised learning and unsupervised learning on several graph datasets.

**Weaknesses:**

* The time cost comparison is missing.
* Deep MMD-GK is more effective than the vanilla MMD-GK but the corresponding algorithm (Algorithm 2) is in the appendix.
* The analysis of the experimental results should be strengthened.

**Questions:**

* Section 3.1 can be made more compact to provide more space for Section 4 and Section 5.
* What is the advantage of the general MMD compared to the single-kernel MMD?
* In Algorithm 1, for the input, the authors mentioned bandwidth. Does this mean the algorithm only use RBF kernels?
* In formula (11), how to determine $S_+$ and $S_-$?
* In Theorem 2, the influence of $n$ and $h$ hasn’t been discussed.
* What is the labeling rate in Table 2?
* In Table 2, on BZR, the values given by supervised and unsupervised Deep MMD-GKs are the same. Is it a typo?
* In Appendix G, are there any explanation for the additional results?

---

> ### Author Response · Authors · 2023-11-22
> **Rebuttal for Reviewer avjd**
>
> **W1:** The time cost comparison is missing.
>
> **Response:**
> Thanks for your comments. We have added the complexity analysis in Section 4.3 of the paper with proof in Appendix G.1. We also show them below.
>
> **Theorem 3.** *The MMD-GK using a kernels $\mathcal{K}$ on a pair of graphs $G$ and $G'$, both with $L$-level of $d$-dimensional node features, can be computed in $O(Lmd+\kappa n^2d)$, where $n$ is the number of nodes, $m$ is the number of edges, and $\kappa$ is the size of the kernel family, i.e. $\kappa=|\mathcal{K}|$. For $N$ graphs, all pairwise MMD-GKs are computed in $O(NLmd+N^2\kappa n^2d)$.*
>
>
> **Theorem 4.** *Suppose the widths of the hidden layers of the neural network are $O(d)$. The Deep MMD-GK using a kernels $\mathcal{K}$ on a pair of graphs $G$ and $G'$, both with $L$-level of $d$-dimensional node features, can be computed in $O(Lnd^2+Lmd+\kappa n^2d)$, where $n$ is the number of nodes, $m$ is the number of edges and $\kappa$ is the size of the kernel family, i.e. $\kappa=|\mathcal{K}|$. For $N$ graphs, all pairwise Deep MMD-GKs are computed in $O(Nnd^2+NLmd+N^2\kappa n^2d)$.*
>
> We compare the theoretical efficiency of the proposed kernel with other baselines in Table 3. Our MMD-GK is competitive with kernels, such as the Shortest Path and Graphlet with a $k>2$. Generally, we set $\kappa$ such that $\kappa d<n$, so the MMD-GK is better in complexity than the Wasserstein Weisfeiler-Lehman graph kernel, when holding $L$ equal to $h$. We note that the Weisfeiler-Lehman Subtree graph kernels scale linearly with the number of nodes; therefore, this method is faster than our approach.
>
> | Graph Kernel                  | Node Labels | Node Attributes |         Complexity         |
> |-------------------------------|:-----------:|:---------------:|:--------------------------:|
> | Graphlet                      |      F      |        F        |          $O(n^k)$          |
> | Shortest Path                 |      T      |        T        |          $O(n^4)$          |
> | Weisfeiler-Lehman Subtree     |      T      |        F        |         $O(hm+hn)$         |
> | Wasserstein Weisfeiler-Lehman |      T      |        T        |    $O(hm + n^3\log(n))$    |
> | MMD-GK                        |      T      |        T        |    $O(Lmd+\kappa n^2d)$    |
> | Deep MMD-GK                   |      T      |        T        | $O(Lnd^2+Lmd+\kappa n^2d)$ |
>
> Above is the summary of selected graph kernels regarding support for node-labeled and node-attributed graphs, and computational complexity. Notations not been introduced yet: $k$: size of largest subgraph considered; $h$: maximum distance between root of neighborhood subgraph/subtree pattern and its nodes.
>
>
> We also add runtimes analysis in Appendix G.2. Following the setting of Togninalli et al. [1], we conducted a simulation using a consistent number of graphs while varying the average node number per graph, to assess the performance of our MMD-GK approach with respect to the average number of nodes. Specifically, we generated 1000 synthetic graphs with the same node feature's dimension $d=20$. For each graph, we generate the number of nodes based on a normal distribution centered on a average node number. For each level of node features $L\in\{1, 2, 3, 4, 5\}$, we applied the MMD-GK with a single Radial Basis Function (RBF) kernel (i.e. $|\mathcal{K}|=1$) to measure and compare runtimes over different average number of nodes $n$.
> As shown by Figure 3 in the Appendix F of the manuscript, the runtime increases with the square of the number of nodes $n^2$, which is modulated by the number of levels $L$.
>
> We also compare our approaches to four graph kernels on the synthetic graphs with node features $d=1$. For the Graphlet kernel, $k=5$; for the Weisfeiler-Lehman Subtree kernel, $h=5$; for our MMD-GK and Deep MMD-GK, $L=h$, $\kappa=1$ and $d=1$. $m$ is held constant across the kerenls. All methods are performed ten times. Refer to the Table 4, we report the total computation time for 10 runs on 1000 synthetic graphs with average of 20 nodes 50 edges, which indicates our approaches are competitive with the existing graph kernel methods.
>
> | Graph Kernel                  | Total Runtime |
> |-------------------------------|:------------:|
> | Graphlet                      |    120.4s    |
> | Shortest Path                 |    30.3.s    |
> | Weisfeiler-Lehman Subtree     |    26.4 s    |
> | Wasserstein Weisfeiler-Lehman |    211.0s    |
> | MMD-GK                        |     29.2s    |
> | Deep MMD-GK                   |     35.2s    |
>
>
> [1] Togninalli, Matteo, et al. "Wasserstein weisfeiler-lehman graph kernels." NeurIPS 2019
>
> ---
>
> **W2:** Deep MMD-GK is more effective than the vanilla MMD-GK but the corresponding algorithm (Algorithm 2) is in the appendix.
>
> **Response:** Thanks for your suggestions. We certainly incorporate your feedback into our manuscript, ensuring that Algorithm of Deep MMD-GK is more accessible by including it in the main body of the text.

---

> > ### Author Response · Authors · 2023-11-22
> > **Further Clarification**
> >
> > **W3:** The analysis of the experimental results should be strengthened.
> >
> > **Response:** Thank you for your feedback on our manuscript. In response, we have updated our discussion in the experiment part.
> >
> > ---
> >
> > **Q1:** Section 3.1 can be made more compact to provide more space for Section 4 and Section 5.
> >
> > **Response:** Thanks for your suggestions. In response, We have condensed this section.
> >
> > **Q2:** What is the advantage of the general MMD compared to the single-kernel MMD?
> >
> > **Response:** Thanks for your inquiry. Note that the kernel family contains the same class of kernel, such as a group of Gaussian kernels with different bandwidths $\{k_{h}: h \in \mathbb{R}_{+}\}$. Compared to the single-kernel MMD, the generalized MMD has the following advantages in practice: 1) auto-search for a suitable and robust bandwidth; 2) avoid the distance collapse as a $k \rightarrow 0$ or $k \rightarrow 1$ leads to $d_k^2 \rightarrow 0$ [1].
> >
> > [1] Fukumizu, Kenji, et al. "Kernel choice and classifiability for RKHS embeddings of probability distributions." Advances in neural information processing systems 22 (2009).
> >
> > **Q3:** In Algorithm 1, for the input, the authors mentioned bandwidth. Does this mean the algorithm only use RBF kernels?
> >
> > **Response:** Thanks for your inquiry. we only adopt RBF kernels for illustration. but the algorithm itself could be extended to other types of kernels. We would consider changing the "bandwidth" into "kernel parameter" to make it more generalized.
> >
> > **Q4:** In formula (11), how to determine $S_{+}$ and  $S_{-}$ ?
> >
> > **Response:** Thanks for your inquiry. We determine the set $S_+$ having the most similar pairs and $S_-$ containing the least similar pairs. Specifically, for each batch of training data $B$, there is a hyparameter $\lambda=0.33$ balancing the size as $|S_-|=|S_+|= \lfloor \lambda |B| \rfloor$.
> >
> > **Q5:** In Theorem 2, the influence of $n$ and $h$ hasn’t been discussed.
> >
> > **Response:** Thanks for bringing up this point. Here is the discussion about $n$ and $h$:
> >
> > -  a larger $h$ implies a smoother kernel that may be more robust to perturbations.
> > -  the deviation in MMD-GK is the most significant when $n = \frac{\epsilon^3} {8(\Delta_{G_1}+\Delta_{G_2})^3}$.
> >     We take the derivative of $\xi$ w.r.t $n$ and observe the critical point:
> >     $\frac{\partial}{\partial n}\left(n(\Delta_{G_1}+\Delta_{G_2})^2 + \frac{\epsilon}{\sqrt{n}}\left(\Delta_{G_1}+\Delta_{G_2}\right)\right) = (\Delta_{G_1}+\Delta_{G_2})^2 -  \frac{\epsilon}{2\sqrt{n}^3}\left(\Delta_{G_1}+\Delta_{G_2}\right)$. As $n \le \frac{\epsilon^3} {8(\Delta_{G_1}+\Delta_{G_2})^3}$, the increase in $n$ leads to a larger $\xi$, which implies that the error due to perturbations becomes larger. And if $n \ge \frac{\epsilon^3} {8(\Delta_{G_1}+\Delta_{G_2})^3}$, the error due to perturbations becomes less significant when $n$ increases.
> >
> > In the revised manuscript, we expand on the implications of $n$ and $h$ in Section 4.1 and Appendix C.1, D.2.
> >
> > **Q6:** What is the labeling rate in Table 2?
> >
> > **Response:** Thanks for the inquiry. If you are referring to the proportion of data that has been labeled out of the total data available, all the datasets have a 100\% labeling rate. If you query about the proportion of labels, please refer to Table below and we have added them into Table 5 in the manuscript.
> >
> > |      Dataset     |   DHFR  |   BZR  |  MUTAG | PTC\_FM | PROTEINS |
> > |:----------------:|:-------:|:------:|:------:|:-------:|:--------:|
> > | Label proportion | 461/295 | 319/86 | 125/63 | 206/143 |  663/450 |
> >
> > **Q7:** In Table 2, on BZR, the values given by supervised and unsupervised Deep MMD-GKs are the same. Is it a typo?
> >
> > **Response:** Thank you for pointing out this concern. Upon re-examination, we found that the values for both supervised and unsupervised Deep MMD-GKs on the BZR dataset are indeed identical (0.910 ± 0.111 for both). This is not a typo error. The identical scores result from the rounding off of values to three decimal places. The actual unrounded scores are slightly different but become identical upon rounding.
> >
> > **Q8:** In Appendix G, are there any explanation for the additional results?
> >
> > **Response:** Thank you for your inquiry. We have added analyses in the appendix of the additional results.
> >
> > **Hope our responses can solve your concerns. We thank the reviewer again for recognizing our work.**

---

> > > ### Comment · Reviewer_avjd · 2023-11-23
> > > **Thanks for your response**
> > >
> > > Thanks for your response! It addressed my concerns, so I'd like to increase my score.

---

> > > > ### Author Response · Authors · 2023-11-23
> > > > **Appreciation for Enhanced Score**
> > > >
> > > > We express our sincere gratitude for upgrading the rating of our paper. The detailed comments and suggestions you provided have undeniably made our paper stronger and more robust. We are truly thankful for the guidance.

---

### Official Review · Reviewer_a9u9 · 2023-10-31

**Soundness:** 2 fair
**Presentation:** 1 poor
**Contribution:** 1 poor
**Rating:** 6
**Confidence:** 3

**Summary:**

This paper discusses graph kernels based on maximum mean discrepancy (MMD) on the (empirical) distribution of embedded nodes. An extension of this MMD graph kernel similar to the deep graph kernel (Yanardag and Vishwanathan 2015) is proposed. For both graph kernel variants the authors provide a theoretical robustness/perturbation analysis. Finally, first experimental results on standard benchmark datasets are presented.

**Strengths:**

* First empirical results on (arguably simple) benchmark datasets are promising.
* Theoretical robustness analysis.

**Weaknesses:**

While the paper is interesting, I vote for rejection in its current form, due the following issues.

Badly written and hard to follow paper:
* The reader is not guided well through the paper.
* While the robustness analyses might be interesting, they are completely unmotivated. The assumptions made and the achieved bounds are hard to interpret and set into context. It is not clear why the authors decided to include them and what the significance of them is. Please guide the reader more and put the results into context, e.g. discuss similar bounds.
* The architecture overview in Fig. 1 is rather confusing and not explained well. Also the "GCN" probably stands for the GCN of Kipf&Welling, however it's not clear what it is doing in the Figure and what is meant e.g. with "vanilla: done. GCN: updating".
* The central definition 6 and Assumption 1 are difficult to understand.


Discussion of related work is insufficient
* Most of the actually related work is not discussed. E.g. many papers exist discussing MMD in the context of graph kernels and GNNs. E.g. [1, 2, Borgwardt's thesis, ... ] Similarly many related work exist that perform graph matching / optimal assignment [Kriege et al., survey, etc.], or modify WL using Wasserstein distances [3-5]. This makes the novelty difficult to assess. Please discuss.

Misleading and wrong claims:
* The authors claim that the proposed kernel has "much lower computational cost" then e.g. the WL kernel. This seems to be wrong. The WL kernel can be computed in near-linear time if the height is fixed. This is similar to the parameter $l$ here in $X^l$. In fact, the computation of WL up to height $l$ should be essentially the same as computing the $X^l$s.
* two of the three main benefits of the proposed approach are (1) dealing with unequal sizes and (2) permutation-invariance. This is however the case for almost all graph kernels, including the popular ones like the WL-kernel, walk kernels, pattern-based kernels etc.
* the claims about degree + power law distributions are unclear. Why should non-power law distributed degrees lead to "inaccurate substructure representations".  Also, the authors claim that GNNs resolve the previously mentioned issue and "capture intricate higher-level interactions". This is however rather misleading as most common GNNs (MPNNs) are bounded in their expressivity by WL and hence cannot capture "more" than the WL kernel (Xu et al., 2019, Morris et al 2019).
* The cited "MMD GAN" paper is mentioned as "MMD also aids in the training of GNN", while the paper actually discussed training of *GANs* and does not deal with graphs at all.

A lot of sentences are vague, handwavy, and unnecessarily wordy, e.g.
* what should "substructures derived from graphs are intertwined" even mean exactly.
* $X^l$ and $X^{(l)}$ are probably the same thing, but the latter is not defined.

Typos and minor issues:
* $\mu_P$ is not defined. Similarly, most readers probably don't know what a characteristic kernel is in the context of MMD. Please make the paper self-contained.
* "to learn graph metric*s*"
* "(2012) . " the space before the dot "."

I am happy to change my score if my concerns are addressed and the contribution/novelty is made clearer.

[1] Ghanem, Hashem et al. "Fast graph kernel with optical random features." ICASSP 2021

[2] O'Bray, Leslie, et al. "Evaluation metrics for graph generative models: Problems, pitfalls, and practical solutions. ICLR 2022

[3] Togninalli, Matteo, et al. "Wasserstein weisfeiler-lehman graph kernels." NeurIPS 2019

[4] Samantha Chen et al. "Weisfeiler-Lehman Meets Gromov-Wasserstein" ICML 2022

[5] Till Schulz et al. "A generalized Weisfeiler-Lehman graph kernel" Machine Learning 2022

**Questions:**

Why do the authors not use the standard unbiased MMD estimator e.g. in Eq. (4)? I.e. the one normalized with $(n\cdot(n-1))$ in the in-distribution sums, cf. Gretton et al. 2012.

---

> ### Author Response · Authors · 2023-11-22
> **Rebuttal for Reviewer a9u9**
>
> **Overview of Responses:**
>
> Dear Reviewer,
>
> Thank you for your insightful feedback. In response to your comments, we have implemented the following enhancements to our manuscript:
>
> 1. **Enhanced Related Work:** We have expanded the discussion in the related work section to better position our study within the existing body of research.
>
> 2. **Elaboration of Motivation and Implications:** We have enriched the Section 3.3 and 4.1 on the motivation behind our robustness analysis.
>
> 3. **Improved Clarity and Readability:** We have carefully revised the manuscript refining the language, streamlining the narrative, and ensuring a more coherent presentation of our research.
>
> We believe these improvements  strengthen the manuscript and look forward to your further feedback.
>
> ---
> **W1:** The reader is not guided well through the paper.
>
> **Response:** Thanks for your comments. In response, we have undertaken a thorough revision to improve the flow and structure.
>
> **W2:** While the robustness analyses might be interesting, they are completely unmotivated. The assumptions made and the achieved bounds are hard to interpret and set into context. It is not clear why the authors decided to include them and what the significance of them is. Please guide the reader more and put the results into context, e.g. discuss similar bounds.
>
> **Response:** Thank you for pointing this out. We have revised this section to explicitly state why these analyses are crucial.
>
> We conduct the robust analysis for the following reasons.
> * First, since real data are often noisy, we would like to theoretically show the upper bound of the gap between the distance of clean graphs and the distance of corrupted graphs. By the way, the robustness is also useful when considering differential privacy, i.e., adding noises to the graphs to protect the data or user privacy in some scenarios.
> * Second, the robust analysis is useful for quantifying the generalization ability of the model. For instance, given a test pair $\tilde{G}_1$ and $\tilde{G}_2$, suppose their closest graphs in the training dataset $\mathcal{G}$ are $G_1, G_2$ respectively. Then according to Theorem 1, $\hat{d}_K^2(\tilde{G}_1^{(l)}, \tilde{G}_2^{(l)}) \leq \hat{d}_K^2(G_1^{(l)}, G_2^{(l)})+\bar{\Delta}$, where $\bar{\Delta}$ is small provided that $G_1, G_2$ are similar to $\tilde{G}_1, \tilde{G}_2$ in terms of the adjacency matrices and node features. A smaller $\bar{\Delta}$ implies a stronger generalization ability.
>
> These discussions have been added to Section 3.2 and Section 4.1.
>
> In the revised paper, we provide more explanation about the assumption (e.g. the norm upper bounds and $\Delta_{A_i}$) in the theorems. They can be found in Section 3.3.
>
> **W3:** The architecture overview in Fig. 1 is rather confusing and not explained well. Also the "GCN" probably stands for the GCN of Kipf&Welling, however it's not clear what it is doing in the Figure and what is meant e.g. with "vanilla: done. GCN: updating".
>
> **Response:** Thanks for your feedback. What we mean by "GCN: updating" is that the deep model requires training and updating parameters, while the vanilla version only goes through the process one time. We changed this figure and its caption to clarify the meaning of terms like "GCN" and "vanilla: done. GCN: updating". Additionally, we will reference the work of [1] to provide context.
>
> [1] Kipf, Thomas N., and Max Welling. "Semi-Supervised Classification with Graph Convolutional Networks." ICLR. 2016.

---

> > ### Author Response · Authors · 2023-11-22
> > **Further Clarification (1)**
> >
> > **W4:** The central definition 6 and Assumption 1 are difficult to understand.
> >
> > **Response:** Thanks for your comments. We acknowledge that Definition 6 and Assumption 1 were not clearly presented. We restructured these sections for clarity, including a more detailed explanation. Here are our explanations in detail.
> >
> > - Definition 6 (Graph Distribution) is a way to understand how the attributes and labels of the nodes in these graphs are spread out or distributed within each group. It's like asking, "If I pick a random graph from the 'Type A' group, what kind of node attributes and labels am I likely to see?" Ideally, graphs in the "Type A" category typically have nodes with certain attributes and labels that are different from those in the "Type B" category. For each category of graphs, there's a pattern or distribution of node attributes and labels. This distribution varies based on the level of $l$. So, when examining the graphs at different levels, we may observe changes in how these attributes and labels are distributed across the nodes.
> >
> > - Assumption 1 is about expecting a certain level of consistency or pattern in the features of graphs that are categorized similarly. Think of a collection of graphs as being grouped based on their labels. What this assumption says is that, within each group of graphs that share the same label, the node features (both attributes and labels) at a specific level don’t just randomly vary. Instead, they follow a specific pattern or distribution. This means that if you pick any graph from this group and look at its node features, these features would be a good representation of the typical features you would expect for this group. For example, $\{X^{(l)}_i: C_i = \mathcal{C}\} \sim \mathbb{P}^{l,\mathcal{C}}_X$ means that the attributes of the nodes in these graphs are distributed in a specific way.
> >
> > **W5:** Most of the actually related work is not discussed...
> >
> > **Response:** Thank you so much for the suggestion. Now, we have included all the papers you mentioned in the revised manuscript, discussed how they relate to our research, and highlighted the novelty of our approach in comparison. They are in Appendix A1 and Appendix A2.  For convenience, we also shown some context of Appendix A2 below:
> >
> > [...] Combining with a graph kernel, [2] obtain the two-sample test for sets of graphs. It is important to note that, [2] just proposed to estimate the MMD for two-sample test with existing graph kernels, rather than to design a graph kernel with an MMD method. When it comes to designing a novel graph kernel, a recent work focusing on speeding up Graphlet kernels, incorporated an MMD metric when showing the effectiveness of their kernel design [3]. However, the kernel itself does not require calculating MMD. Instead, MMD is only used to demonstrate the classification power of their embeddings.
> >
> > Recent research have integrated MMD into the realm of GNNs. One key application is in the comparison of graph-structured data to evaluate generated graphs [4,5], for instance, when determining if two graphs exhibit similar structural properties or when comparing the node embeddings generated by GNNs from different graphs. MMD also aids in the training of GNNs [6], ensuring that the distribution of the generated node embeddings aligns well with the target distribution. This can be especially crucial in semi-supervised settings where labeled data is sparse but there's a need to ensure that the graph's overall structure is well captured. Recently, [7] critically evaluated the use of MMD in graph generative model comparison, proposing practical recommendations for its effective application.
> >
> > **In our approach, we introduce a novel graph kernel that uniquely integrates the concept of Maximum Mean Discrepancy (MMD) with Graph Neural Networks (GNNs), differing from existing efforts that either apply MMD for two-sample tests using traditional graph kernels or use MMD to evaluate the performance of GNNs without directly incorporating it into the kernel design.**
> >
> > ---
> >
> > [2] Borgwardt, Karsten Michael. "Graph kernels." PhD diss., 2007.
> >
> > [3] Ghanem, Hashem et al. "Fast graph kernel with optical random features." ICASSP 2021
> >
> > [4] Dai, Hanjun, et al. "Scalable deep generative modeling for sparse graphs." PMLR, 2020.
> >
> > [5] Chen, Xiaohui, et al. "Order Matters: Probabilistic Modeling of Node Sequence for Graph Generation." PMLR, 2021.
> >
> > [6] Roncoli, Andrea, et al. "Domain Adaptive Graph Neural Networks for Constraining Cosmological Parameters Across Multiple Data Sets."
> >
> > [7] O'Bray, Leslie, et al. "Evaluation metrics for graph generative models: Problems, pitfalls, and practical solutions. ICLR 2022

---

> > > ### Author Response · Authors · 2023-11-22
> > > **Further Clarification (2)**
> > >
> > > **W6:** The authors claim that the proposed kernel has "much lower computational cost" then e.g. the WL kernel. This seems to be wrong. The WL kernel can be computed in near-linear time if the height is fixed. This is similar to the parameter $l$ here in $X^{l}$ . In fact, the computation of WL up to height should be essentially the same as computing the $X^{l}$s.
> > >
> > >  **Response:** Thank you for pointing this out. We corrected this claim by adding a complexity analysis part, offering a more accurate comparison. Here are some discussion of the complexity.
> > >
> > > **Theorem 3.** *The MMD-GK using a kernels $\mathcal{K}$ on a pair of graphs $G$ and $G'$, both with $L$-level of $d$-dimensional node features, can be computed in $O(Lmd+\kappa n^2d)$, where $n$ is the number of nodes, $m$ is the number of edges, and $\kappa$ is the size of the kernel family, i.e. $\kappa=|\mathcal{K}|$. For $N$ graphs, all pairwise MMD-GKs are computed in $O(NLmd+N^2\kappa n^2d)$.*
> > >
> > >
> > > **Theorem 4.** *Suppose the widths of the hidden layers of the neural network are $O(d)$. The Deep MMD-GK using a kernels $\mathcal{K}$ on a pair of graphs $G$ and $G'$, both with $L$-level of $d$-dimensional node features, can be computed in $O(Lnd^2+Lmd+\kappa n^2d)$, where $n$ is the number of nodes, $m$ is the number of edges and $\kappa$ is the size of the kernel family, i.e. $\kappa=|\mathcal{K}|$. For $N$ graphs, all pairwise Deep MMD-GKs are computed in $O(Nnd^2+NLmd+N^2\kappa n^2d)$.*
> > >
> > > We compare the theoretical efficiency of the proposed kernel with other baselines in Table 3. Our approaches are as efficient as classical kernels and is more efficient than the Wasserstein WL kernel. We note that the Weisfeiler-Lehman Subtree graph kernels scale linearly with the number of nodes; therefore, this method is faster than our approach.
> > >
> > > | Graph Kernel                  | Node Labels | Node Attributes |         Complexity         |
> > > |-------------------------------|:-----------:|:---------------:|:--------------------------:|
> > > | Graphlet                      |      F      |        F        |          $O(n^k)$          |
> > > | Shortest Path                 |      T      |        T        |          $O(n^4)$          |
> > > | Weisfeiler-Lehman Subtree     |      T      |        F        |         $O(hm+hn)$         |
> > > | Wasserstein Weisfeiler-Lehman |      T      |        T        |    $O(hm + n^3\log(n))$    |
> > > | MMD-GK                        |      T      |        T        |    $O(Lmd+\kappa n^2d)$    |
> > > | Deep MMD-GK                   |      T      |        T        | $O(Lnd^2+Lmd+\kappa n^2d)$ |
> > >
> > > Above is the summary of selected graph kernels regarding support for node-labeled and node-attributed graphs, and computational complexity. Notations not been introduced yet: $k$: size of largest subgraph considered; $h$: maximum distance between root of neighborhood subgraph/subtree pattern and its nodes.
> > >
> > >
> > > We also add runtimes analysis in Appendix G.2. Following the setting of [8], we conducted a simulation using a consistent number of graphs while varying the average node number per graph, to assess the performance of our MMD-GK approach with respect to the average number of nodes. Specifically, we generated 1000 synthetic graphs with the same node feature's dimension $d=20$. For each graph, we generate the number of nodes based on a normal distribution centered on a average node number. For each level of node features $L\in\{1, 2, 3, 4, 5\}$, we applied the MMD-GK with a single Radial Basis Function (RBF) kernel (i.e. $|\mathcal{K}|=1$) to measure and compare runtimes over different average number of nodes $n$. As shown by Figure 3 in the Appendix G of the manuscript, the runtime increases with the square of the number of nodes $n^2$, which is modulated by the number of levels $L$.
> > >
> > > Moreover, we compare our approaches to four graph kernels on the synthetic graphs with node features $d=1$. For the Graphlet kernel, $k=5$; for the Weisfeiler-Lehman Subtree kernel, $h=5$; for our MMD-GK and Deep MMD-GK, $L=h$, $\kappa=1$ and $d=1$. $m$ is held constant across the kerenls. All methods are performed ten times. As the table shows below, we report the total computation time for 10 runs on 1000 synthetic graphs with average of 20 nodes 50 edges, which indicates our approaches are competitive with the existing graph kernel methods.
> > >
> > > | Graph Kernel                  | Total Runtime |
> > > |-------------------------------|:------------:|
> > > | Graphlet                      |    120.4s    |
> > > | Shortest Path                 |    30.3.s    |
> > > | Weisfeiler-Lehman Subtree     |    26.4 s    |
> > > | Wasserstein Weisfeiler-Lehman |    211.0s    |
> > > | MMD-GK                        |     29.2s    |
> > > | Deep MMD-GK                   |     35.2s    |
> > >
> > >
> > > [8] Togninalli, Matteo, et al. "Wasserstein weisfeiler-lehman graph kernels." NeurIPS 2019

---

> > > > ### Author Response · Authors · 2023-11-22
> > > > **Further Clarification (3)**
> > > >
> > > > **W7:** two of the three main benefits of the proposed approach are (1) dealing with unequal sizes and (2) permutation-invariance. This is however the case for almost all graph kernels, including the popular ones like the WL-kernel, walk kernels, pattern-based kernels etc.
> > > >
> > > > **Response:** Thank you for your insightful review. Indeed, the ability to handle graphs of unequal sizes and permutation invariance are fundamental features of many graph kernel methods, our proposed approach builds upon these foundational benefits and aims to enhance them in a generalized form of graph data. Specifically, we are interested in graphs with either node attributes or labels, or both.
> > > >
> > > > **W8:** the claims about degree + power law distributions are unclear. Why should non-power law distributed degrees lead to "inaccurate substructure representations". Also, the authors claim that GNNs resolve the previously mentioned issue and "capture intricate higher-level interactions". This is however rather misleading as most common GNNs (MPNNs) are bounded in their expressivity by WL and hence cannot capture "more" than the WL kernel (Xu et al., 2019, Morris et al 2019).
> > > >
> > > > **Response:** Thank you for raising these points. If the subtree is of depth zero (i.e., only one root vertex), we can represent it using the vertex degree. However, the vertex degree distribution of a graph does not always follow a power-law distribution. Thus, the learned representations for substructures are not accurate. In light of your feedback, the revised paper will remove the statement about power-law degree to avoid confusion. Also we add claims about the capabilities of GNNs in comparison to the WL kernel, avoiding overstatements about their expressivity and instead focusing on the ways they process and integrate node features for graph-related tasks.
> > > >
> > > > **W9:** The cited "MMD GAN" paper is mentioned as "MMD also aids in the training of GNN", while the paper actually discussed training of GANs and does not deal with graphs at all.
> > > >
> > > > **Response:** We apologize for the incorrect citation and replace it with the right one. We hope this correction will ensure that our references are relevant and accurately support our claims.
> > > >
> > > > **W10:** what should "substructures derived from graphs are intertwined" even mean exactly.
> > > >
> > > > **Response:** Thank you for your feedback. In the context of the paper, 'intertwined substructures' refer to the occurrence of subgraphs that share nodes or edges, leading to overlapping features within the generated feature map. To address the confusion, we will ensure that the revised paper explains this concept clearly.
> > > >
> > > > **W11:** $X^{l}$ and $X^{(l)}$ are probably the same thing, but the latter is not defined.
> > > >
> > > > **Response:** Thanks for your comments. We have updated $X^{(l)}$ in Definition 2 and ensured consistency in our notation throughout the paper to avoid any confusion.
> > > >
> > > > **W12:** $\mu_P$ is not defined. Similarly, most readers probably don't know what a characteristic kernel is in the context of MMD. Please make the paper self-contained."to learn graph metrics"; "(2012) . " the space before the dot "."
> > > >
> > > > **Response:** We have addressed each typo and minor issue you pointed out, including defining $\mu_{P}$ and providing a clearer explanation of what a characteristic kernel is in the context of MMD. Additionally, we corrected the formatting errors and strive to make the paper self-contained and comprehensible to a broader audience.
> > > >
> > > > ---
> > > >
> > > > **Q1:** Why do the authors not use the standard unbiased MMD estimator e.g. in Eq. (4)? i.e. the one normalized with $(n\cdot(n-1))$ in the in-distribution sums, cf. Gretton et al. 2012.
> > > >
> > > > **Response:** Thanks for your inquery, we acknowledge that the unbiased MMD $MMD_{u}^2$ is produced by normalizing with $(n\cdot(n-1))$ in the in-distribution sums. However, the reason why we do not use this estimator is that $MMD_{u}^2$ may be negative, as highlighted in [9]. To clarify, when $d^2_{\mathcal{K}}$ calculates a similarity measure, it ideally should fall within the range of [0,1] according to Definition 7 (MMD between two graphs). This range makes the interpretation more straightforward. In our experiments, we found that using both biased and unbiased estimators yielded almost identical outcomes, further supporting our decision to opt for the estimator we used.
> > > >
> > > > ---
> > > >
> > > > [9] Gretton, Arthur, et al. "A kernel two-sample test." JMLR. 2012.

---

> > > > > ### Comment · Reviewer_a9u9 · 2023-11-23
> > > > >
> > > > > Thank you for these very thorough clarifications! I raised my score.

---

> > > > > > ### Author Response · Authors · 2023-11-23
> > > > > > **Appreciation for Enhanced Score**
> > > > > >
> > > > > > We express our sincere gratitude for upgrading the rating of our paper. The detailed comments and suggestions you provided have undeniably made our paper stronger and more robust. We are truly thankful for the guidance.

---

### Official Review · Reviewer_9SJd · 2023-11-03

**Soundness:** 3 good
**Presentation:** 4 excellent
**Contribution:** 4 excellent
**Rating:** 8
**Confidence:** 4

**Summary:**

This paper proposes a new graph kernel to calculate the maximum mean discrepancy (MMD) which measures the similarity of two graphs. The main idea is to regard each graph as a sampling distribution from a latent metric space where a similarity kernel can be designed for efficiency. Based on such an MMD-based graph kernel, two methods are designed for unsupervised and supervised graph metric learning, respectively.

**Strengths:**

+ The paper is well-written, and I find it a joy to read. The authors present a nice and coherent story to introduce the current gap and motivation of the idea.

+ The proposed MMD-based kernels are technically sound and will have a significant impact on the graph metric learning domain.

+ In-depth and well-organized theoretical analysis is provided to justify the approach in a principled manner.

**Weaknesses:**

- The major limitation is the lack of theoretical and empirical study of the efficiency of the proposed kernel. As highlighted in the introduction the proposed method aims to address the computation and memory cost of existing methods, no evidence shows this goal is achieved.

**Questions:**

See weaknesses.

---

> ### Author Response · Authors · 2023-11-22
> **Rebuttal for Reviewer 9SJd**
>
> **W1:** The major limitation is the lack of theoretical and empirical study of the efficiency of the proposed kernel. As highlighted in the introduction the proposed method aims to address the computation and memory cost of existing methods, no evidence shows this goal is achieved.
>
> **Response:**
> Thanks for your comments. We have added the complexity analysis in Section 4.3 of the paper with proof in Appendix G.1. We also show them below.
>
> **Theorem 3.** *The MMD-GK using a kernels $\mathcal{K}$ on a pair of graphs $G$ and $G'$, both with $L$-level of $d$-dimensional node features, can be computed in $O(Lmd+\kappa n^2d)$, where $n$ is the number of nodes, $m$ is the number of edges, and $\kappa$ is the size of the kernel family, i.e. $\kappa=|\mathcal{K}|$. For $N$ graphs, all pairwise MMD-GKs are computed in $O(NLmd+N^2\kappa n^2d)$.*
>
>
> **Theorem 4.** *Suppose the widths of the hidden layers of the neural network are $O(d)$. The Deep MMD-GK using a kernels $\mathcal{K}$ on a pair of graphs $G$ and $G'$, both with $L$-level of $d$-dimensional node features, can be computed in $O(Lnd^2+Lmd+\kappa n^2d)$, where $n$ is the number of nodes, $m$ is the number of edges and $\kappa$ is the size of the kernel family, i.e. $\kappa=|\mathcal{K}|$. For $N$ graphs, all pairwise Deep MMD-GKs are computed in $O(Nnd^2+NLmd+N^2\kappa n^2d)$.*
>
> We compare the theoretical efficiency of the proposed kernel with other baselines in Table 3. Our MMD-GK is competitive with kernels, such as the Shortest Path and Graphlet with a $k>2$. Generally, we set $\kappa$ such that $\kappa d<n$, so the MMD-GK is better in complexity than the Wasserstein Weisfeiler-Lehman graph kernel, when holding $L$ equal to $h$. We note that the Weisfeiler-Lehman Subtree graph kernels scale linearly with the number of nodes; therefore, this method is faster than our approach.
>
> | Graph Kernel                  | Node Labels | Node Attributes |         Complexity         |
> |-------------------------------|:-----------:|:---------------:|:--------------------------:|
> | Graphlet                      |      F      |        F        |          $O(n^k)$          |
> | Shortest Path                 |      T      |        T        |          $O(n^4)$          |
> | Weisfeiler-Lehman Subtree     |      T      |        F        |         $O(hm+hn)$         |
> | Wasserstein Weisfeiler-Lehman |      T      |        T        |    $O(hm + n^3\log(n))$    |
> | MMD-GK                        |      T      |        T        |    $O(Lmd+\kappa n^2d)$    |
> | Deep MMD-GK                   |      T      |        T        | $O(Lnd^2+Lmd+\kappa n^2d)$ |
>
> Above is the summary of selected graph kernels regarding support for node-labeled and node-attributed graphs, and computational complexity. Notations not been introduced yet: $k$: size of largest subgraph considered; $h$: maximum distance between root of neighborhood subgraph/subtree pattern and its nodes.
>
>
> In addition, we add runtimes analysis in Appendix G.2. Following the setting of Togninalli et al. [1], we conducted a simulation using a consistent number of graphs while varying the average node number per graph, to assess the performance of our MMD-GK approach with respect to the average number of nodes. Specifically, we generated 1000 synthetic graphs with the same node feature's dimension $d=20$. For each graph, we generate the number of nodes based on a normal distribution centered on a average node number. For each level of node features $L\in\{1, 2, 3, 4, 5\}$, we applied the MMD-GK with a single Radial Basis Function (RBF) kernel (i.e. $|\mathcal{K}|=1$) to measure and compare runtimes over different average number of nodes $n$.
> As shown by Figure 3 in the Appendix F of the manuscript, the runtime increases with the square of the number of nodes $n^2$, which is modulated by the number of levels $L$.
>
> We also compare our approaches to four graph kernels on the synthetic graphs with node features $d=1$. For the Graphlet kernel, $k=5$; for the Weisfeiler-Lehman Subtree kernel, $h=5$; for our MMD-GK and Deep MMD-GK, $L=h$, $\kappa=1$ and $d=1$. $m$ is held constant across the kerenls. All methods are performed ten times. Refer to the Table 4, we report the total computation time for 10 runs on 1000 synthetic graphs with average of 20 nodes 50 edges, which indicates our approaches are competitive with the existing graph kernel methods.
>
> | Graph Kernel                  | Total Runtime |
> |-------------------------------|:------------:|
> | Graphlet                      |    120.4s    |
> | Shortest Path                 |    30.3.s    |
> | Weisfeiler-Lehman Subtree     |    26.4 s    |
> | Wasserstein Weisfeiler-Lehman |    211.0s    |
> | MMD-GK                        |     29.2s    |
> | Deep MMD-GK                   |     35.2s    |
>
> [1] Togninalli, Matteo, et al. "Wasserstein weisfeiler-lehman graph kernels." NeurIPS 2019
>
> **Hope this response can solve your concerns. We thank the reviewer again for recognizing our work.**

---

### Author Response · Authors · 2023-11-22
**Global Response**

Compared to the original submission, we made the following major changes in the revised paper:

1. Added the theoretical computational complexity (Theorems 3 and 4) and the corresponding comparison (Table 1).

2. Supplemented the time cost comparisons (Table 4 in Appendix G).

3. Added Wasserstein WL kernel to the experiments of graph classification (Table 4).

4. Added two GNN baselines to the experiments of graph clustering (Table 3).

5. Provided more discussion on the motivation and implication of the robustness analysis (Section 3.3 and 4.1).

6. Provided more intuition about the loss functions (Section 4).

7. The explicit formulation for the stability parameter $\omega$ is provided in Appendix F.

8. Polished the writing and improved the readability.

---

### Meta-Review · Area_Chair_NzTS · 2023-12-05

**Metareview:**

This paper propose a novel way to construct graph kernels via maximum mean discrepancy (MMD) that is more general and computational efficient. The paper also shows its connection to existing graph comparison metric as well as empirical performances in various graph learning tasks. The author-reviewer discussion generate additional insight to further improve the paper in many aspects, which has been reflected in the revised manuscript. The results would be very beneficial for the community to learn and understand and I would suggest to accept.

**Justification For Why Not Higher Score:**

Despite all the plus point, the clarity can be further improved.

**Justification For Why Not Lower Score:**

Given the interesting idea, various connections to existing works, and extensive empirical experiments, the results from the paper deserve descent attention from our ICLR audience so I suggest spotlight accept.

---

### Decision · Program_Chairs · 2024-01-16

Accept (spotlight)